# ER-to-lysosome Ca²⁺ refilling followed by K⁺ efflux-coupled store-operated Ca²⁺ entry in inflammasome activation and metabolic inflammation

**Hyereen Kang[1†], Seong Woo Choi[2†], Joo Young Kim[3], Soo-Jin Oh[4], Sung Joon Kim[5]\*, Myung-Shik Lee[1,4]\***

[1]Severance Biomedical Science Institute, Yonsei University College of Medicine, Seoul, Republic of Korea; [2]Department of Physiology and Ion Channel Disease Research Center, Dongguk University College of Medicine, Gyeongju, Republic of Korea; [3]Department of Pharmacology and Brain Korea 21 Project for Medical Sciences, Yonsei University College of Medicine, Seoul, Republic of Korea; [4]Soonchunhyang Institute of Medi-bio Science and Division of Endocrinology, Department of Internal Medicine, Soonchunhyang University College of Medicine, Cheonan, Republic of Korea; [5]Department of Physiology, Ischemic/Hypoxic Disease Institute, Seoul National University College of Medicine, Seoul, Republic of Korea

**\*For correspondence:**
physiolksj@gmail.com (SJK);
mslee0923@sch.ac.kr (M-SL)

[†]These authors contributed equally to this work

**Abstract** We studied lysosomal Ca²⁺ in inflammasome. Lipopolysaccharide (LPS) + palmitic acid (PA) decreased lysosomal Ca²⁺ ($[Ca^{2+}]_{Lys}$) and increased $[Ca^{2+}]_i$ through mitochondrial ROS, which was suppressed in *Trpm2*-KO macrophages. Inflammasome activation and metabolic inflammation in adipose tissue of high-fat diet (HFD)-fed mice were ameliorated by *Trpm2* KO. ER→lysosome Ca²⁺ refilling occurred after lysosomal Ca²⁺ release whose blockade attenuated LPS + PA-induced inflammasome. Subsequently, store-operated Ca²⁺entry (SOCE) was activated whose inhibition suppressed inflammasome. SOCE was coupled with K⁺ efflux whose inhibition reduced ER Ca²⁺ content ($[Ca^{2+}]_{ER}$) and impaired $[Ca^{2+}]_{Lys}$ recovery. LPS + PA activated KCa3.1 channel, a Ca²⁺-activated K⁺ channel. Inhibitors of KCa3.1 channel or *Kcnn4* KO reduced $[Ca^{2+}]_{ER}$, attenuated increase of $[Ca^{2+}]_i$ or inflammasome activation by LPS + PA, and ameliorated HFD-induced inflammasome or metabolic inflammation. Lysosomal Ca²⁺ release induced delayed JNK and ASC phosphorylation through CAMKII-ASK1. These results suggest a novel role of lysosomal Ca²⁺ release sustained by ER→lysosome Ca²⁺ refilling and K⁺ efflux through KCa3.1 channel in inflammasome activation and metabolic inflammation.

## eLife assessment

This **useful** study proposes a role of lysosomal Ca²⁺ release in inflammasome signaling and metabolic inflammation. While the proposed model would be of considerable interest to the field of immunology if validated, the experimental approaches to study calcium dynamics are problematic, with one of several concerns being the transfection efficiency. The major claims of the article are thus only **incompletely** supported.

## Introduction

Lysosomotropic agents are classical inflammasome activators (*Hornung et al., 2008*; *Misawa et al., 2013*; *Xian et al., 2022*). The detailed mechanism of inflammasome activation by lysosomal stress has been unclear, while roles of lysosomal $Ca^{2+}$ release were suggested (*Okada et al., 2014*). $K^+$ efflux is crucial in most inflammasome activations, facilitating NLRP3-NEK7 oligomerization (*He et al., 2016*; *Sharif et al., 2019*). The relationship between $K^+$ efflux and $Ca^{2+}$ flux in inflammasome has hardly been studied, despite their potential link (*Yaron et al., 2015*).

We studied lysosomal events in inflammasome focusing on the roles of lysosomal $Ca^{2+}$ release by lipopolysaccharide (LPS) + palmitic acid (PA), an effector of metabolic stress (*Nakamura et al., 2009*) (LP). We found that LP induces mitochondrial reactive oxygen species (ROS) that activates a lysosomal $Ca^{2+}$ efflux channel (TRPM2), leading to lysosomal $Ca^{2+}$ release, delayed JNK activation, ASC phosphorylation, and inflammasome activation. We also found occurrence of ER→lysosome $Ca^{2+}$ refilling sustaining lysosomal $Ca^{2+}$ efflux and subsequent store-operated $Ca^{2+}$ entry (SOCE) in inflammasome. Finally, we elucidated the roles of $K^+$ efflux facilitating SOCE through hyperpolarization-accelerated extracellular $Ca^{2+}$ influx (*Guéguinou et al., 2014*), and identified KCa3.1 $Ca^{2+}$-activated $K^+$ channel as the $K^+$ efflux channel in LP-induced inflammasome and metabolic inflammation.

## Results

### Lysosomal $Ca^{2+}$ release by mitochondrial ROS in inflammasome

We investigated whether lysosomal $Ca^{2+}$ release occurs in inflammasome activation by LP, a combination activating inflammasome related to metabolic inflammation (*Wen et al., 2011*). When we studied perilysosomal $Ca^{2+}$ release in bone marrow-derived macrophages (MΦs) (BMDMs) transfected with GCaMP3-ML1 (*Shen et al., 2012*), lysosomal $Ca^{2+}$ release was not directly visualized by LP; however, perilysosomal $Ca^{2+}$ release by Gly-Phe β-naphthylamide (GPN), a lysosomotropic agent (*Shen et al., 2012*), was significantly reduced (*Figure 1A*, *Figure 1—source data 1*), suggesting preemptying or release of lysosomal $Ca^{2+}$ by LP, similar to the results using other inducers of lysosomal $Ca^{2+}$ release (*Park et al., 2022*; *Zhang et al., 2016*). We next measured lysosomal $Ca^{2+}$ content ($[Ca^{2+}]_{Lys}$) that can be affected by lysosomal $Ca^{2+}$ release. $[Ca^{2+}]_{Lys}$ determined using Oregon Green BAPTA-1 Dextran (OGBD) was significantly lowered by LP (*Figure 1B*, *Figure 1—source data 1*), consistent with lysosomal $Ca^{2+}$ release. Likely due to lysosomal $Ca^{2+}$ release, $[Ca^{2+}]_i$ measured by Fluo-3-AM staining was significantly increased by LP (*Figure 1C*, *Figure 1—source data 1*). Ratiometric $[Ca^{2+}]_i$ measurement using Fura-2 to avoid uneven loading or quenching validated $[Ca^{2+}]_i$ increase by LP (*Figure 1C*, *Figure 1—source data 1*). Functional roles of increased $[Ca^{2+}]_i$ in inflammasome by LP were revealed by abrogation of IL-1β release or IL-1β maturation by BAPTA-AM, a cell-permeable $Ca^{2+}$ chelator (*Figure 1D and E*, *Figure 1—source data 1 and 2*). Immunoblotting (IB) demonstrated that pro-IL-1β level was not notably affected by BAPTA-AM, suggesting that the effect of BAPTA-AM was unrelated to the potential inhibition of pro-IL-1β transcription or translation (*Figure 1E*, *Figure 1—source data 2*).

We next studied the mechanism of lysosomal $Ca^{2+}$ release by LP. Since several (lysosomal) ion channels can be activated by ROS (*Zhang et al., 2016*) and ROS can be produced by PA due to mitochondrial complex inhibition (*Nakamura et al., 2009*), we studied ROS accumulation. LP effectively induced CM-H2DCFDA fluorescence indicating ROS accumulation, while PA alone or LPS alone induced only a little ROS accumulation (*Figure 1—figure supplement 1A*, *Figure 1—figure supplement 1—source data 1*), suggesting synergistic effect of LPS and PA. When we studied the roles of ROS in $[Ca^{2+}]_i$ increase using *N*-acetyl cysteine (NAC), an antioxidant (*Tardiolo et al., 2018*), release and maturation of IL-1β by LP were significantly reduced (*Figure 1—figure supplement 1B*, *Figure 1E*, *Figure 1—source data 2*, *Figure 1—figure supplement 1—source data 1*). NAC abrogated LP-induced increase of $[Ca^{2+}]_i$ as well (*Figure 1—figure supplement 1C*, *Figure 1—figure supplement 1—source data 1*), substantiating the functional roles of ROS in $[Ca^{2+}]_i$ increase and inflammasome activation. We also studied mitochondrial ROS because mitochondria is a well-known target of PA, an effector of metabolic stress (*Nakamura et al., 2009*) and critical in inflammasome (*Xian et al., 2022*; *Zhou et al., 2011*). Using MitoSOX, we observed significant mitochondrial ROS accumulation by LP. When we quenched mitochondrial ROS using MitoTEMPOL (*Figure 1—figure supplement 1D*, *Figure 1—figure supplement 1—source data 1*), IL-1β release by LP was significantly reduced

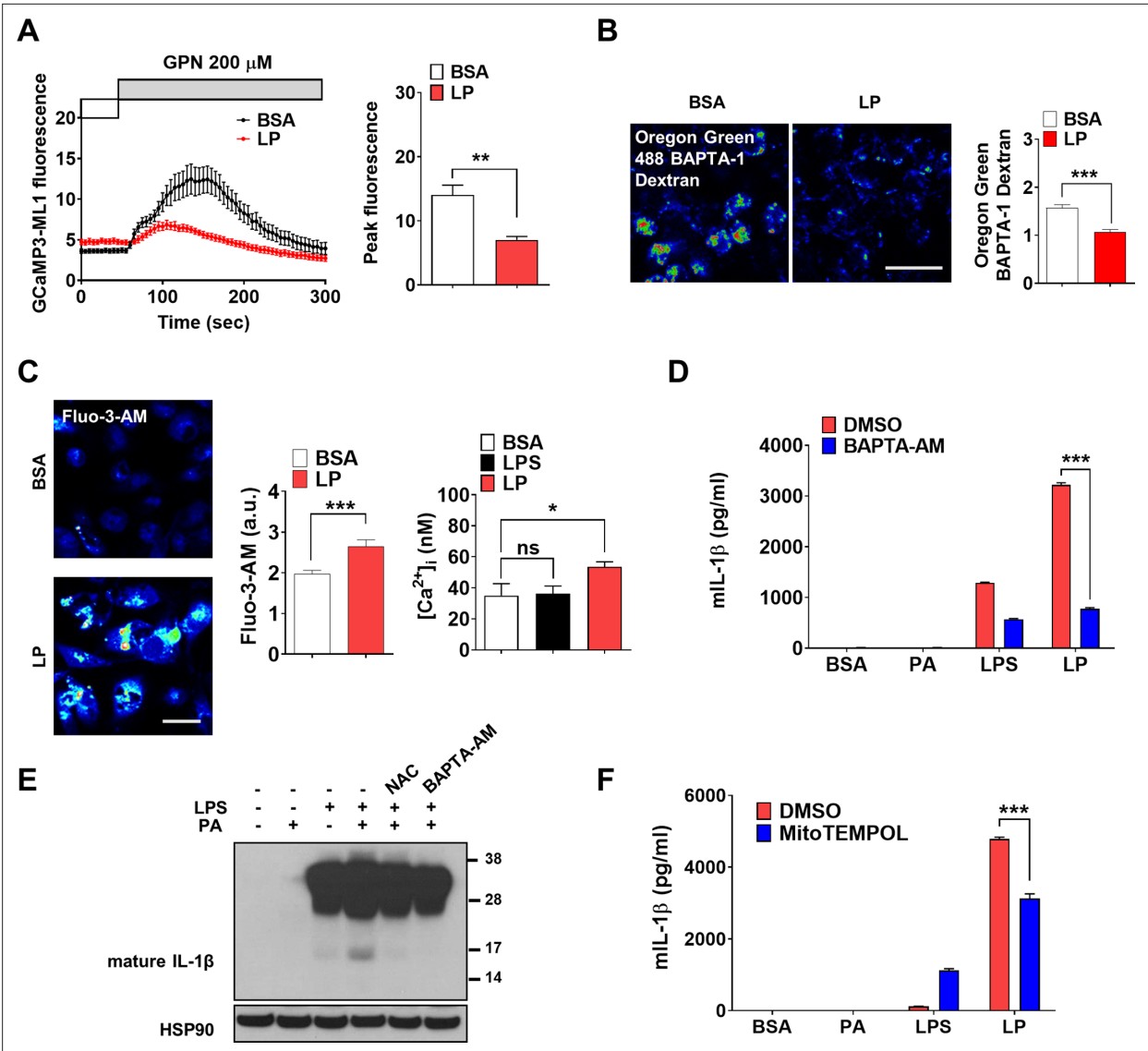

**Figure 1.** Lysosomal $Ca^{2+}$ and mitochondrial reactive oxygen species (ROS) in inflammasome. (**A**) Perilysosomal fluorescence after applying GPN to GCaMP3-ML1-transfected bone marrow-derived macrophages (BMDMs) treated with LP for a total of 4 hr including lipopolysaccharide (LPS) pretreatment for 3 hr (left) (actual LP treatment time is 1 hr). Peak fluorescence (right) (n = 6). (**B**) $[Ca^{2+}]_{Lys}$ in Oregon Green BAPTA-1 Dextran (OGBD)-loaded MΦs treated with LP for a total of 4 hr including LPS pretreatment for 3 hr (right). Representative fluorescence images (left) (n = 8). (**C**) $[Ca^{2+}]_i$ in MΦs treated with LP for a total of 4 hr including LPS pretreatment for 3 hr, determined using Fluo-3-AM staining (middle) or Fura-2 (right). Representative Fluo-3 images (left) (n = 7 for BSA; n = 6 for LPS; n = 13 for LP). (**D**) IL-1β ELISA of culture supernatant after treating peritoneal MΦs with LPS alone or palmitic acid (PA) alone for 21 hr, or with LP for a total of 21 hr including LPS pretreatment for 3 hr in the presence or absence of BAPTA-AM (n = 3). (**E**) Immunoblotting (IB) of lysate of peritoneal MΦs treated with LPS alone or PA alone for 21 hr, or with LP for a total of 21 hr including LPS pretreatment for 3 hr in the presence or absence of BAPTA-AM or *N*-acetyl cysteine (NAC), using indicated Abs. (**F**) IL-1β ELISA of culture supernatant after treating MΦs with LPS alone or PA alone for 21 hr, or with LP for a total of 21 hr including LPS pretreatment for 3 hr in the presence or absence of MitoTEMPOL (n = 3). Data shown as means ± SEM from more than three independent experiments. *p<0.05, **p<0.01, and ***p<0.001 by two-tailed Student's *t*-test (**A, B**), one-way ANOVA with Tukey's test (**C**), or two-way ANOVA with Sidak test (**D, F**) (ns, not significant). Scale bars, 20 μm.

The online version of this article includes the following source data and figure supplement(s) for figure 1:

**Source data 1.** Original data for fluorescence and ELISA.

**Source data 2.** Original uncropped blot.

**Figure supplement 1.** Mitochondrial reactive oxygen species (ROS) in inflammasome activation by lipopolysaccharide + palmitic acid (LPS + PA) (LP).

**Figure supplement 1—source data 1.** Original data for fluorescence and ELISA.

(*Figure 1F*, *Figure 1—source data 1*), suggesting crucial roles of mitochondrial ROS in LP-induced inflammasome.

## Ca²⁺ release through lysosomal TRPM2 channel in inflammasome

We next studied which lysosomal Ca²⁺ exit channel is involved in lysosomal Ca²⁺ release by LP. Previous papers suggested the roles of transient receptor potential melastatin 2 (TRPM2) channel on the plasma membrane, in inflammasome by other stimulators (*Tseng et al., 2017*; *Wang et al., 2020*). We hypothesized that TRPM2 on lysosome could be involved in LP-induced inflammasome since TRPM2 has been reported to be expressed on lysosome as well (*Sumoza-Toledo and Penner, 2011*) and *Trpm2*-KO mice are resistant to diet-induced glucose intolerance (*Zhang et al., 2012*). We verified the expression of TRPM2 on lysosome of BMDMs by colocalization of TRPM2 and LAMP2 (*Figure 2—figure supplement 1A*). While the physiological ligand of TRPM2 is ADP-ribose (ADPR), ROS can activate TRPM2 by increasing ADPR production from poly-ADPR (*Wang et al., 2020*). We thus studied the effects of apigenin or quercetin inhibiting ADPR generation through CD38 inhibition (*Bock, 2020*; *Nam et al., 2020*). Inflammasome by LP was significantly inhibited by apigenin or quercetin (*Figure 2—figure supplement 1B*, *Figure 2—figure supplement 1—source data 1*), suggesting that TRPM2 participates in inflammasome by LP. [Ca²⁺]ᵢ increase by LP was also reduced by apigenin or quercetin (*Figure 2—figure supplement 1C*, *Figure 2—figure supplement 1—source data 1*), supporting that apigenin or quercetin suppresses inflammasome through the inhibition of lysosomal Ca²⁺ channels activated by ADPR such as TRPM2. Since apigenin or quercetin has effects other than CD38 inhibition (*Li et al., 2016*; *Zhang et al., 2020*), we employed *Trpm2*-KO mice. [Ca²⁺]ᵢ increase and [Ca²⁺]ₗᵧₛ decrease by LP were abrogated in *Trpm2*-KO MΦs (*Figure 2A and B*, *Figure 2—source data 1*), suggesting the roles of TRPM2 in lysosomal Ca²⁺ release by LP. However, ROS production by LP was not changed by *Trpm2* KO (*Figure 2—figure supplement 1D*, *Figure 2—figure supplement 1—source data 1*), suggesting that TRPM2 is downstream of ROS. Importantly, inflammasome activation assessed by IL-1β ELISA of culture supernatant or IB of peritoneal MΦ lysate after LP treatment was significantly reduced by *Trpm2* KO (*Figure 2C*, *Figure 2—source data 1 and 2*), indicating the roles of TRPM2, likely lysosomal TRPM2, in inflammasome by LP. ASC speck formation, a marker of inflammasome, by LP was also markedly reduced by *Trpm2* KO (*Figure 2D*, *Figure 2—source data 1*). When we employed other activators, inflammasome activation by L-Leucyl-L-Leucine methyl ester (LLOMe) or monosodium urate (MSU), lysosomotropic agents, was significantly inhibited by *Trpm2* KO. However, inflammasome by nigericin, an ionophore exchanging K⁺ for H⁺ or ATP acting on P2X7 ion channel on the plasma membrane (*Campden and Zhang, 2019*), was not significantly affected (*Figure 2—figure supplement 1E*, *Figure 2—figure supplement 1—source data 1*), suggesting that TRPM2 channel is crucial in inflammasome involving lysosomal Ca²⁺.

Despite its crucial role in inflammasome by LP, TRPM2 exists on both plasma membrane and lysosome. We thus studied whether plasma membrane TRPM2 current can be activated by LP employing N-(p-amylcinnamoyl)anthranilic acid (ACA), an inhibitor of TRPM2 (*Kraft et al., 2006*). ACA could inhibit IL-1β release by LP as expected (*Figure 3—figure supplement 1A*). Intracellular dialysis using cyclic ADPR induced slow inward current on the plasma membrane, which was inhibited by ACA (*Figure 3—figure supplement 1B*, *Figure 3—figure supplement 1—source data 1*). The amplitude of cyclic ADPR-induced inward current was not affected by PA and/or LPS (b–c in *Figure 3—figure supplement 1B and C*, *Figure 3—figure supplement 1—source data 1*). Basal inward current inhibited by ACA, that is, unstimulated TRPM2 activity, was also not changed by PA and/or LPS (a–c in *Figure 3—figure supplement 1B and C*, *Figure 3—figure supplement 1—source data 1*), suggesting that LP neither affects nor induces plasma membrane TRPM2 current and that lysosomal TRPM2 is likely important in LP-induced inflammasome. To further study the roles of lysosomal TRPM2 in inflammasome activation by LP, we employed bafilomycin A1 emptying lysosomal Ca²⁺ reservoir through lysosomal v-ATPase inhibition (*Kinnear et al., 2004*; *Lange et al., 2009*). TRPM2-dependent [Ca²⁺]ᵢ increase by LP was abrogated by bafilomycin A1 (*Figure 3—figure supplement 1D*, *Figure 3—figure supplement 1—source data 1*), strongly supporting Ca²⁺ release through lysosomal TRPM2 by LP.

## Ameliorated metabolic inflammation by *Trpm2* KO

Since lysosomal TRPM2 is likely involved in inflammasome by LP, we next studied the roles of TRPM2 in inflammasome and metabolic inflammation in vivo. Nonfasting blood glucose was significantly

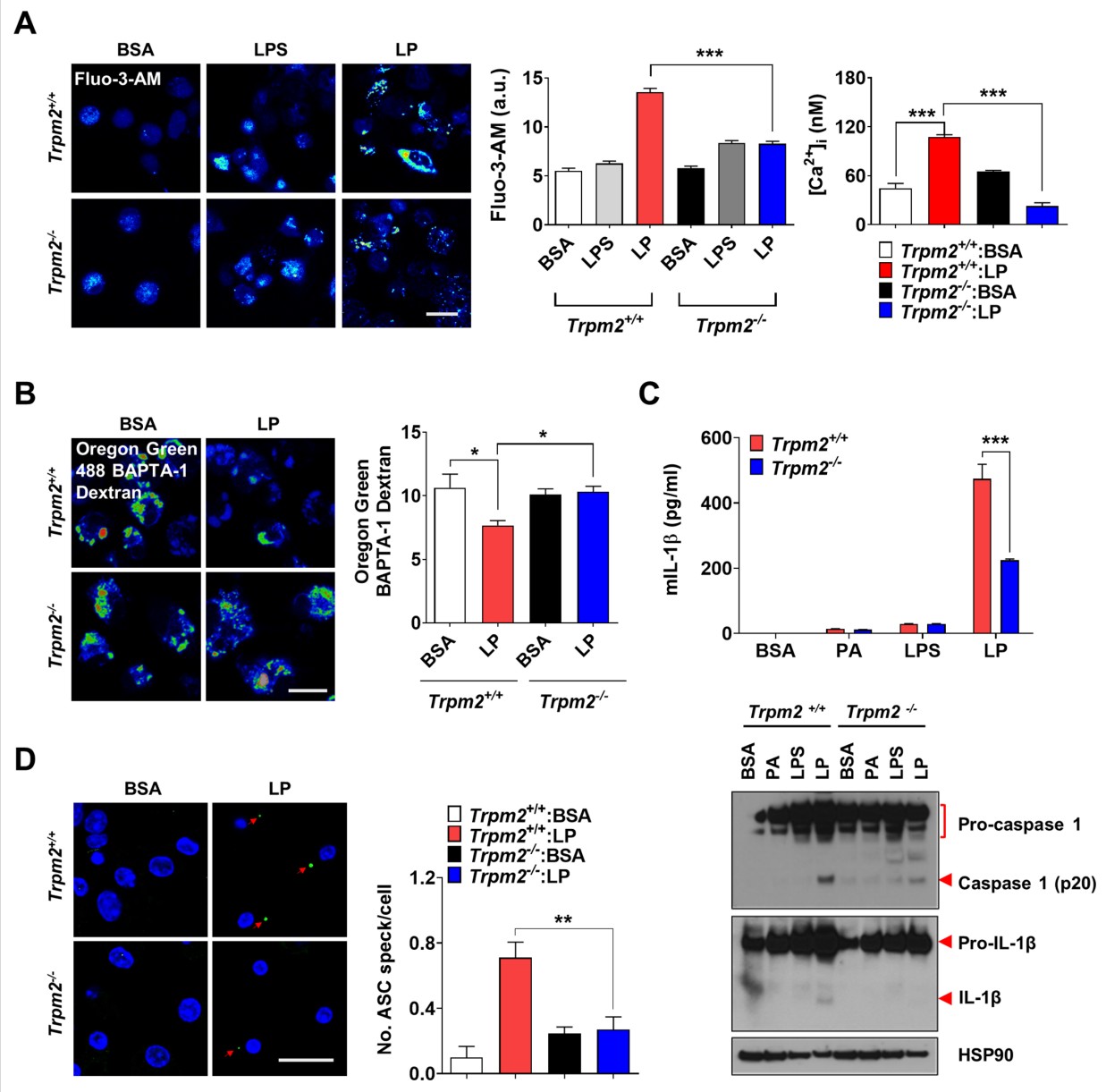

**Figure 2.** Lysosomal Ca$^{2+}$ efflux through TRPM2 in inflammasome. (**A**) [Ca$^{2+}$]$_i$ in peritoneal MΦs treated with lipopolysaccharide (LPS) alone for 4 hr or with LP for a total of 4 hr including LPS pretreatment for 3 hr, determined using Fluo-3-AM (middle) or Fura-2 (right). Representative Fluo-3 images (left) (n = 8 for Fluo-3-AM; n = 9 for Fura-2). (**B**) [Ca$^{2+}$]$_{Lys}$ in MΦs treated with LP for a total of 4 hr including LPS pretreatment for 3 hr, determined by Oregon Green BAPTA-1 Dextran (OGBD) loading (right). Representative fluorescence images (left) (n = 5) (**C**) IL-1β ELISA of culture supernatant (upper) and immunoblotting (IB) of cell lysate using indicated Abs after treatment of MΦs with LPS alone for 21 hr or LP for a total of 21 hr including LPS pretreatment for 3 (lower). (n = 4) (**D**) The number of ASC specks in MΦs treated with LP for a total of 21 hr including LPS pretreatment for 3 hr, determined by immunofluorescence using anti-ASC Ab (right). Representative confocal images (left) (n = 4 for *Trpm2*$^{+/+}$:BSA; n = 5 for *Trpm2*$^{+/+}$: LP; n = 5 for *Trpm2*$^{-/-}$:BSA; n = 7 for *Trpm2*$^{-/-}$:LP) Data shown as means ± SEM from more than three independent experiments. *p<0.05, **p<0.01, and ***p<0.001 by one-way ANOVA with Tukey's test (**A, B, D**), or two-way ANOVA with Sidak test or Bonferroni test (**C**). Scale bars, 20 μm.

The online version of this article includes the following source data and figure supplement(s) for figure 2:

**Source data 1.** Original data for fluorescence and ELISA.

**Source data 2.** Original uncropped blot.

**Figure supplement 1.** Effect of CD38 inhibitors and *Trpm2* KO on inflammasome.

**Figure supplement 1—source data 1.** Original data for fluorescence and ELISA.

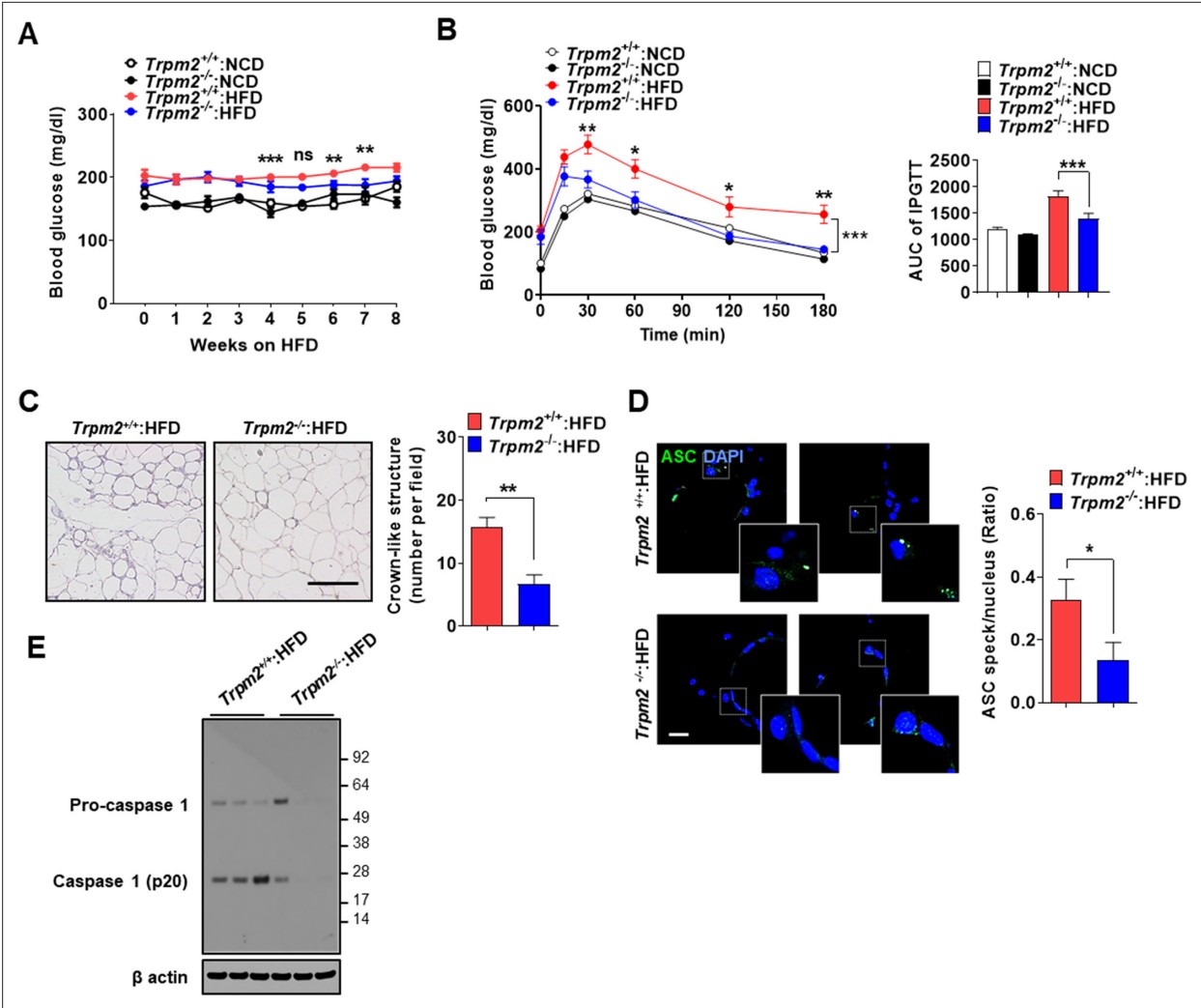

**Figure 3.** Ameliorated metabolic inflammation by *Trpm2* KO. (**A**) Nonfasting blood glucose of mice on normal chow diet (NCD) (n = 5 each) or high-fat diet (HFD) (n = 8 each). (*comparison between *Trpm2⁺/⁺* and *Trpm2⁻/⁻* mice on HFD). (**B**) Intraperitoneal glucose tolerance test (IPGTT) after NCD (n = 5 each) or HFD (n = 8 each) feeding for 8 weeks (left). Area under the curve (AUC) (right). (**C**) The number of crown-like structures (CLS) in while adipose tissue (WAT) after HFD feeding for 8 weeks (right). Representative H&E sections (left) (scale bar, 50 μm) (n = 8 each). (**D**) The number of ASC specks in WAT after HFD feeding for 8 weeks, determined by immunofluorescence using anti-ASC Ab (right). Representative confocal images (left) (scale bar, 20 μm) (insets, magnified) (n = 7 each). (**E**) Immunoblotting (IB) of stromal vascular fraction (SVF) of WAT after HFD feeding for 8 weeks using indicated Abs (n = 3). Data shown as means ± SEM from more than three independent experiments. *p<0.05, **p<0.01, and ***p<0.001 by two-way ANOVA (**B**) or two-tailed Student's *t*-test (**A, C, D**).

The online version of this article includes the following source data and figure supplement(s) for figure 3:

**Source data 1.** Original data for blood glucose, and the number of CLS and ASC specks.

**Source data 2.** Original uncropped blot.

**Figure supplement 1.** Plasma membrane TRPM2 current in MΦs and metabolic profile of *Trpm2*-KO mice.

**Figure supplement 1—source data 1.** Original data for current, [Ca²⁺]ᵢ, body weight, and HOMA-IR.

lower in *Trpm2*-KO mice on high-fat diet (HFD) compared to control mice on HFD (***Figure 3A***, ***Figure 3—source data 1***), while body weight was not different between them (***Figure 3—figure supplement 1E***, ***Figure 3—figure supplement 1—source data 1***). Intraperitoneal glucose tolerance test (IPGTT) showed significantly ameliorated glucose intolerance and reduced area under the curve (AUC) in *Trpm2*-KO mice on HFD (***Figure 3B***, ***Figure 3—source data 1***). HOMA-IR, an index of insulin resistance, was also significantly reduced in *Trpm2*-KO mice on HFD (***Figure 3—figure supplement 1F***, ***Figure 3—figure supplement 1—source data 1***).

We studied whether improved metabolic profile by *Trpm2* KO is due to reduced inflammasome. The number of crown-like structures (CLS) representing metabolic inflammation (*Weisberg et al., 2003*) was significantly reduced in while adipose tissue (WAT) of *Trpm2*-KO mice on HFD (*Figure 3C*, *Figure 3—source data 1*), suggesting reduced metabolic inflammation by *Trpm2* KO. Inflammasome activation was also significantly reduced in WAT of *Trpm2*-KO mice on HFD as evidenced by the significantly reduced numbers of ASC specks and capase-1 cleavage (*Figure 3D and E*, *Figure 3—source data 1 and 2*), indicating that TRPM2 is important in inflammasome related to metabolic syndrome.

## ER→lysosome Ca²⁺ refilling in inflammasome

After confirming the roles of lysosomal TRPM2 in $[Ca^{2+}]_i$ increase by LP, we studied the changes in ER $Ca^{2+}$ content ($[Ca^{2+}]_{ER}$) as ER, the largest intracellular $Ca^{2+}$ reservoir, interacts with other organelles in cellular processes requiring intracellular $Ca^{2+}$ flux. While $[Ca^{2+}]_{Lys}$ is comparable to $[Ca^{2+}]_{ER}$ (*Raffaello et al., 2016*), lysosome alone might not be a major $Ca^{2+}$ source because of small volume (*Penny et al., 2014*). Thus, we studied whether ER to lysosomal $Ca^{2+}$ flux, which has been observed after lysosomal $Ca^{2+}$ emptying (*Garrity et al., 2016*; *Park et al., 2022*), occurs during inflammasome activation to sustain lysosomal $Ca^{2+}$ release. When we measured $[Ca^{2+}]_{ER}$ in GEM-CEPIA1er (*Suzuki et al., 2014*)-transfected BMDMs treated with LP without extracellular $Ca^{2+}$ to abolish SOCE (*Derler et al., 2016*), $[Ca^{2+}]_{ER}$ became significantly lower (*Figure 4A*, *Figure 4—source data 1*), suggesting $Ca^{2+}$ flux from ER to lysosome likely to replenish reduced $[Ca^{2+}]_{Lys}$ during inflammasome. $[Ca^{2+}]_{ER}$ determined using a FRET-based D1ER (*Park et al., 2009*) also demonstrated a significantly reduced $[Ca^{2+}]_{ER}$ by LP in a $Ca^{2+}$-free KRB buffer (*Figure 4A*, *Figure 4—source data 1*). To study the dynamic changes of $[Ca^{2+}]_{ER}$ and its temporal relationship with $[Ca^{2+}]_{Lys}$, we simultaneously traced $[Ca^{2+}]_{ER}$ and $[Ca^{2+}]_{Lys}$ in GEM-CEPIA1er-transfected cells loaded with OGBD. When $[Ca^{2+}]$ was monitored in cells that have reduced $[Ca^{2+}]_{Lys}$ after LP treatment and then were incubated without LP in a $Ca^{2+}$-free medium, recovery of decreased $[Ca^{2+}]_{Lys}$ was noted (*Figure 4B*, *Figure 4—source data 1*). In this condition, $[Ca^{2+}]_{ER}$ decrease occurred in parallel with $[Ca^{2+}]_{Lys}$ recovery (*Figure 4B*, *Figure 4—source data 1*), strongly indicating ER→lysosome $Ca^{2+}$ refilling.

We next studied which ER $Ca^{2+}$ exit channels are involved in ER→lysosome $Ca^{2+}$ refilling. When LP was removed after treatment, $[Ca^{2+}]_{Lys}$ recovery was observed in a $Ca^{2+}$-replete medium (*Figure 4C*, *Figure 4—source data 1*). Here, Xestospongin C, an $IP_3$ receptor ($IP_3R$) channel antagonist (*Garrity et al., 2016*), inhibited recovery of $[Ca^{2+}]_{Lys}$ after LP removal (*Figure 4C*, *Figure 4—source data 1*), suggesting ER→lysosome $Ca^{2+}$ refilling through $IP_3R$ channel. Dantrolene, an antagonist of ryanodine receptor (RyR) channel, another ER $Ca^{2+}$ exit channel (*Garrity et al., 2016*), did not significantly affect recovery of $[Ca^{2+}]_{Lys}$ (*Figure 4C*, *Figure 4—source data 1*). When we chelated ER $Ca^{2+}$ with a membrane-permeant metal chelator *N,N,N',N'*-tetrakis (2-pyridylmethyl)ethylene diamine (TPEN) that has a low $Ca^{2+}$ affinity and can chelate ER $Ca^{2+}$ but not cytosolic $Ca^{2+}$ (*Hofer et al., 1998*), recovery of $[Ca^{2+}]_{Lys}$ after LP removal was markedly inhibited (*Figure 4C*, *Figure 4—source data 1*), again supporting ER→lysosome $Ca^{2+}$ refilling during lysosomal $Ca^2$ recovery. When functional impact of ER→lysosome $Ca^{2+}$ refilling was studied, Xestospongin C significantly suppressed IL-1β release by LP (*Figure 4D*, *Figure 4—source data 1*), suggesting that ER→lysosome $Ca^{2+}$ refilling contributes to inflammasome by LP. Since ER→lysosome $Ca^{2+}$ refilling could be facilitated by membrane contact between organelles (*Yang et al., 2019*), we studied apposition of ER and lysosome proteins. Proximity ligation assay (PLA) demonstrated multiple contact between VAPA on ER and ORP1L on lysosome by LP (*Figure 4E*, *Figure 4—source data 1*), suggesting the facilitation of ER→lysosome $Ca^{2+}$ refilling by organelle contact.

As extracellular $Ca^{2+}$ is likely to enter cells through SOCE channel after ER $Ca^{2+}$ depletion (*Derler et al., 2016*), we next studied extracellular $Ca^{2+}$. When extracellular $Ca^{2+}$ was chelated by 3 mM EGTA reducing $[Ca^{2+}]_i$ in RPMI medium to 99 nM (*Schoenmakers et al., 1992*) below $[Ca^{2+}]_i$ in $Ca^{2+}$-free medium (*Maggi et al., 1989*), IL-1β release by LP was significantly reduced (*Figure 4—figure supplement 1A*, *Figure 4—figure supplement 1—source data 1*), demonstrating the roles of extracellular $Ca^{2+}$ in inflammasome by LP. 2-APB, a SOCE inhibitor, also significantly reduced IL-1β release by LP (*Figure 4—figure supplement 1B*, *Figure 4—figure supplement 1—source data 1*), supporting the roles of SOCE in inflammasome. Since 2-APB can inhibit ER $Ca^{2+}$ channel as well, we next employed another SOCE inhibitor. BTP2, a SOCE inhibitor that does not affect ER $Ca^{2+}$ channel (*Zitt et al., 2004*), significantly reduced IL-1β release by LP (*Figure 4—figure supplement 1C*, *Figure 4—figure*

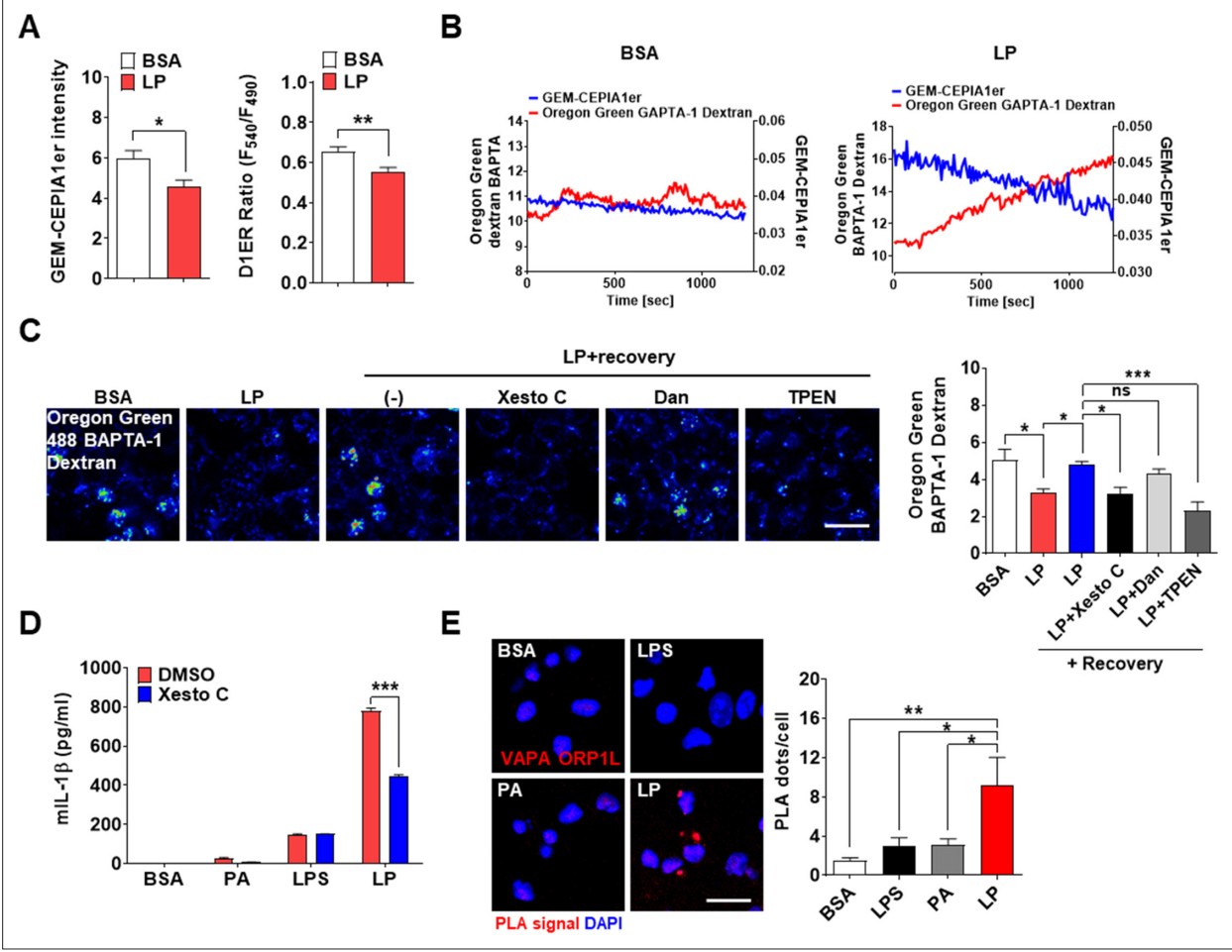

**Figure 4.** ER→lysosome $Ca^{2+}$ refilling in inflammasome. (**A**) $[Ca^{2+}]_{ER}$ in GEM-CEPIA1er- (left) or D1ER-transfected bone marrow-derived macrophages (BMDMs) (right) treated with LP for 1 hr without extracellular $Ca^{2+}$ after lipopolysaccharide (LPS) pretreatment for 3 hr (n = 26 for BSA; n = 25 for LP). (**B**) BMDMs transfected with GEM-CEPIA1er and loaded with Oregon Green BAPTA-1 Dextran (OGBD) were treated with LP for 1 hr after LPS pretreatment for 3 hr (right) or BSA alone for 4 hr (left). Tracing of $[Ca^{2+}]_{Lys}$ and $[Ca^{2+}]_{ER}$ after change to a fresh medium without extracellular $Ca^{2+}$ (n = 4 for BSA; n = 4 for LP). (**C**) OGBD-loaded BMDMs were treated with LP for 1 hr after LPS pretreatment for 3 hr. Recovery of $[Ca^{2+}]_{Lys}$ after change to a fresh medium with or without Xestospongin C (Xesto C), dantrolene (Dan), or TPEN (right). Representative confocal images (left) (n = 9 for BSA; n = 8 for LP; n = 9 for LP + Recovery; n = 6 for LP + Recovery + Xesto C; n = 5 for LP + Recovery + Dan; n = 8 for LP + Recovery + TPEN). (**D**) IL-1β ELISA of culture supernatant after treating peritoneal MΦs with LPS alone or palmitic acid (PA) alone for 21 hr, or with LP for a total of 21 hr including LPS pretreatment for 3 hr, in the presence or absence of Xesto C (n = 4). (**E**) PLA in BMDMs treated with LPS alone or PA alone for 21 hr, or LP for a total of 21 hr including LPS pretreatment for 3 hr, using Abs to VAPA and ORP1L (right). Representative fluorescence images (left) (n = 4). Data shown as means ± SEM from more than three independent experiments. *p<0.05, **p<0.01, and ***p<0.001 by two-tailed Student's *t*-test (**A**), one-way ANOVA with Tukey's test (**C, E**) or two-way ANOVA with Sidak test (**D**). Scale bar, 20 μm.

The online version of this article includes the following source data and figure supplement(s) for figure 4:

**Source data 1.** Original data for fluorescence (ratio), ELISA, and the number of PLT dots.

**Figure supplement 1.** Store-operated $Ca^{2+}$ entry (SOCE) in inflammasome by LP.

**Figure supplement 1—source data 1.** Original data for ELISA and fluorescence ratio.

*supplement 1—source data 1*), substantiating the roles of SOCE in inflammasome. We also studied whether BTP2 can affect $[Ca^{2+}]_{ER}$ in LP-induced inflammasome through SOCE inhibition. We determined $[Ca^{2+}]_{ER}$ without extracellular $Ca^{2+}$ removal because BTP2 effect on SOCE cannot be seen after extracellular $Ca^{2+}$ removal abrogating SOCE. In the presence of extracellular $Ca^{2+}$, $[Ca^{2+}]_{ER}$ was not decreased by LP, likely due to SOCE (*Figure 4—figure supplement 1D*, *Figure 4—figure supplement 1—source data 1*). Here, BTP2 significantly reduced $[Ca^{2+}]_{ER}$ of LP-treated BMDMs (*Figure 4—figure supplement 1D*, *Figure 4—figure supplement 1—source data 1*) likely due to SOCE inhibition, suggesting that SOCE is activated in inflammasome by LP to replenish reduced ER $Ca^{2+}$ store. We also

studied aggregation of STIM1, a $Ca^{2+}$ sensor in ER, which can be observed in SOCE activation (*Derler et al., 2016*). Indeed, STIM1 aggregation was clearly observed after LP treatment (*Figure 4—figure supplement 1E*), which was colocalized with ORAI1, an SOCE channel on the plasma membrane (*Vaca, 2010*), strongly indicating SOCE through ORAI1 channel in LP-induced inflammasome.

## Coupling of K⁺ efflux and Ca²⁺ influx in inflammasome

$K^+$ efflux is one of the most common and critical events in inflammasome (*Muñoz-Planillo et al., 2013*), although a couple of inflammasomes without $K^+$ efflux have been reported (*Groß et al., 2016*). In excitable cells, $K^+$ efflux leads to hyperpolarization, which negatively modulates $Ca^{2+}$ influx through voltage-gated $Ca^{2+}$ channel activated by depolarization. In nonexcitable cells, contrarily, $Ca^{2+}$ influx though voltage-independent $Ca^{2+}$ channel such as SOCE channel can be positively modulated by $K^+$ efflux due to increased electrical $Ca^{2+}$ driving force (*Guéguinou et al., 2014*). Since $Ca^{2+}$ influx from extracellular space into cytosol might be positively modulated by $K^+$ efflux in nonexcitable cells such as MΦs, we studied whether $K^+$ efflux is coupled to $Ca^{2+}$ influx. Intracellular $K^+$ content ($[K^+]_i$) determined using Potassium Green-2-AM was decreased by LP (*Figure 5—figure supplement 1A*, *Figure 5—figure supplement 1—source data 1*), likely due to $K^+$ efflux. High extracellular $K^+$ content ($[K^+]_e$) (60 mM) inhibited inflammasome activation by LP likely by inhibiting $K^+$ efflux (*Figure 5—figure supplement 1B*, *Figure 5—figure supplement 1—source data 1*). We then studied the effects of high $[K^+]_e$ on $[Ca^{2+}]_{ER}$ that would be affected by high $[K^+]_e$ if $K^+$ efflux and $Ca^{2+}$ influx through SOCE are coupled. We determined $[Ca^{2+}]_{ER}$ again without extracellular $Ca^{2+}$ removal since the possible effects of high $[K^+]_e$ on SOCE cannot be seen after extracellular $Ca^{2+}$ removal. At $[K^+]_e$ of 5.4 mM ($[K^+]_e$ in RPMI), $[Ca^{2+}]_{ER}$ was not decreased by LP. However, at high $[K^+]_e$, LP significantly reduced $[Ca^{2+}]_{ER}$ (*Figure 5A*, *Figure 5—source data 1*) likely due to SOCE inhibition by high $[K^+]_e$, suggesting that $K^+$ efflux is coupled to extracellular $Ca^{2+}$ influx through SOCE. We next studied whether $[Ca^{2+}]_{Lys}$ recovery, which is seen after LP removal due to ER→lysosome $Ca^{2+}$ refilling, is affected by high $[K^+]_e$. $[Ca^{2+}]_{Lys}$ recovery after LP removal became significantly lower by high $[K^+]_e$ (*Figure 5B*, *Figure 5—source data 1*), suggesting that high $[K^+]_e$ dampens $[Ca^{2+}]_{Lys}$ recovery after LP removal likely through SOCE inhibition. High $[K^+]_e$ also suppressed $[Ca^{2+}]_i$ increase by LP (*Figure 5C*, *Figure 5—source data 1*), suggesting the contribution of $K^+$ efflux in $[Ca^{2+}]_i$ increase by LP through sustained $Ca^{2+}$ influx via SOCE.

We next investigated which $K^+$ efflux channel is involved in inflammasome, focusing on $Ca^{2+}$-activated $K^+$ channels that can positively modulate $Ca^{2+}$ influx after initial $Ca^{2+}$ entry (*Guéguinou et al., 2014*). When MΦs were incubated with various $K^+$ efflux channel inhibitors, charybdotoxin (CTX) inhibiting all three types of $Ca^{2+}$-activated $K^+$ channels [BK, IKCa1 (KCa3.1), and SK] (*Chen and Chung, 2013*; *González et al., 2012*) significantly suppressed IL-1β release by LP (*Figure 5—figure supplement 1C*, *Figure 5—figure supplement 1—source data 1*), supporting the roles of $Ca^{2+}$-activated $K^+$ channels in inflammasome. In contrast, inhibitors of $K^+$ efflux channels unrelated to $Ca^{2+}$-induced activation such as quinine (a 2-pore $K^+$ channel inhibitor), barium sulfate (Kir channel inhibitor), or 4-aminopyridine (4-AP, a Kv channel inhibitor) did not significantly inhibit IL-1β release by LP (*Figure 5—figure supplement 1D*, *Figure 5—figure supplement 1—source data 1*). We next studied which channels among $Ca^{2+}$-activated $K^+$ channels are involved. Paxillin, a BK channel inhibitor (*González et al., 2012*), and UCL 1684 or apamin, SK channel inhibitors (*Chen et al., 2021*; *Strøbaek et al., 2000*), did not significantly affect IL-1β release by LP (*Figure 5—figure supplement 1D*, *Figure 5—figure supplement 1—source data 1*). In contrast, TRAM-34, a specific IKCa1 (KCa3.1) channel inhibitor (*Agarwal et al., 2013*; *Wulff et al., 2000*), significantly decreased IL-1β release by LP (*Figure 5—figure supplement 1C and D*, *Figure 5—figure supplement 1—source data 1*), suggesting the involvement of KCa3.1. We next studied the effects of TRAM-34 on $[Ca^{2+}]_{ER}$. While $[Ca^{2+}]_{ER}$ decrease by LP was not seen without extracellular $Ca^{2+}$ removal likely due to SOCE, it was clearly observed when TRAM-34 was added (*Figure 5—figure supplement 1E*, *Figure 5—figure supplement 1—source data 1*), consistent with the positive roles of KCa3.1 in $Ca^{2+}$ influx after ER $Ca^{2+}$ emptying. $[Ca^{2+}]_i$ increase by LP was also inhibited by TRAM-34 (*Figure 5D*, *Figure 5—source data 1*), demonstrating the contribution of KCa3.1 channel in the increase of $[Ca^{2+}]_i$. Further, TRAM-34 abrogated decrease of $[K^+]_i$ by LP (*Figure 5E*, *Figure 5—source data 1*), indicating the roles of KCa3.1 in $K^+$ efflux during inflammasome activation. BTP2 also suppressed increase of $[Ca^{2+}]_i$ and decrease of $[K^+]_i$ by LP (*Figure 5D and E*, *Figure 5—source data 1*), likely by inhibiting $Ca^{2+}$ influx that can activate $Ca^{2+}$-activated $K^+$ channel such as KCa3.1.

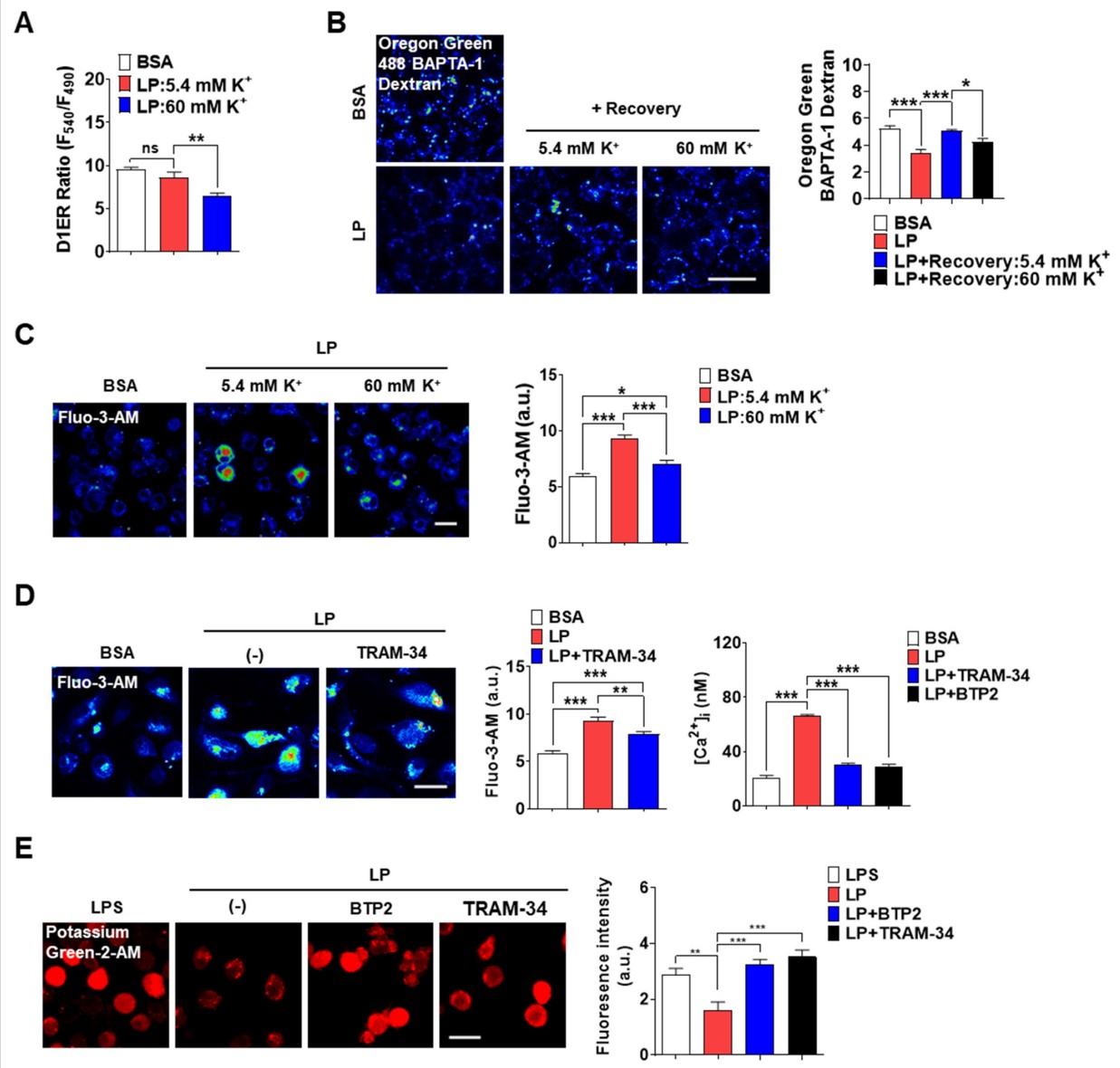

**Figure 5.** Coupling of K$^+$ efflux and Ca$^{2+}$ influx in inflammasome. (**A**) [Ca$^{2+}$]$_{ER}$ in D1ER-transfected bone marrow-derived macrophages (BMDMs) treated with LP for a total of 4 hr including lipopolysaccharide (LPS) pretreatment for 3 hr at [K$^+$]$_e$ of 5.4 or 60 mM (n = 21 each). (**B**) Oregon Green BAPTA-1 Dextran (OGBD)-loaded MΦs were treated with LP for a total of 4 hr including LPS pretreatment for 3 hr. Recovery of [Ca$^{2+}$]$_{Lys}$ in a fresh medium with 5.4 or 60 mM K$^+$ (right). Representative fluorescence images (left) (n = 7 for BSA; n = 6 for LP; n = 7 for LP + Recovery:5.4 mM K$^+$; n = 6 for LP + Recovery:60 mM K$^+$). (**C**) [Ca$^{2+}$]$_i$ in MΦs treated with LP for a total of 4 hr including after LPS pretreatment for 3 hr in a medium with 5.4 or 60 mM K$^+$ (right). Representative Fluo-3 images (left) (n = 14 for BSA; n = 8 for LP:5.4 mM K$^+$; n = 6 for LP:60 mM K$^+$). (**D**) [Ca$^{2+}$]$_i$ in MΦs treated with LP for 1 hr in the presence or absence of BTP2 or TRAM-34 after LPS pretreatment for 3 hr, determined using Fluo-3-AM (middle) or Fura-2 (right). Representative Fluo-3 images (left) (for Fluo-3-AM, n = 8) (for Fura-2, n = 13). (**E**) [K$^+$]$_i$ after LP treatment for a total of 21 hr including LPS pretreatment for 3 hr in the presence or absence of BTP2 or TRAM-34 (right). Representative Potassium Green-2 images (left) (n = 5). Data shown as means ± SEM from more than three independent experiments. *p<0.05, **p<0.01, and ***p<0.001 by one-way ANOVA with Tukey's test (**A–E**). Scale bar, 20 µm.

The online version of this article includes the following source data and figure supplement(s) for figure 5:

**Source data 1.** Original data for fluorescence (ratio) and [Ca$^{2+}$]$_i$.

**Figure supplement 1.** Effect of high extracellular K$^+$ and inhibitors of K$^+$ efflux channels on inflammasome.

**Figure supplement 1—source data 1.** Original data for fluorescence (ratio) and [Ca$^{2+}$]$_i$.

We next studied whether KCa3.1 K$^+$ current is activated by LP. If Ca$^{2+}$-activated K$^+$ channel coupled to Ca$^{2+}$ influx mediates K$^+$ efflux in inflammasome activation, conventional whole-cell patch clamp using Ca$^{2+}$-clamped solution would nullify the effects of Ca$^{2+}$ influx, rendering the study of increased Ca$^{2+}$ influx impossible. Hence, we employed nystatin-perforated patch clamp technique that leaves elevated [Ca$^{2+}$]$_i$ intact, as nystatin pores are permeable to monovalent but not to divalent ions (*Akaike and Harata, 1994*). When ramp-like voltage clamp was applied to induce brief current-voltage (I/V) curves and TRAM-34-inhibitable current was obtained by digital subtraction, slope of I/V curves sensitive to TRAM-34 was increased by LP but not by LPS alone (*Figure 6A and B*, *Figure 6—source data 1*), suggesting increased KCa3.1 activity by LP. Expression of *Kcnn4* was not significantly affected by LPS and/or PA (*Figure 6—figure supplement 1A*, *Figure 6—figure supplement 1—source data 1*), suggesting that increased KCa3.1 activity by LP is not due to *Kcnn4* induction. When we employed *Kcnn4*-KO MΦs that do not express *Kcnn4* (*Figure 6—figure supplement 1B*, *Figure 6—figure supplement 1—source data 1*), TRAM-34-inhibitable K$^+$ current was not observed before or after LP treatment (*Figure 6C and D*, *Figure 6—source data 1*), confirming the reliability of results obtained by nystatin-perforated patch clamp technique.

We next studied the functional roles of *Kcnn4* in inflammasome. IL-1β release and inflammasome activation by LP were significantly reduced by *Kcnn4* KO (*Figure 6E*, *Figure 6—source data 1 and 2*), supporting that KCa3.1 channel is important in LP-induced inflammasome. TNF-α or IL-6 release by LPS or LP was not significantly changed by *Kcnn4* KO (*Figure 6—figure supplement 1C*, *Figure 6—figure supplement 1—source data 1*). We also studied the effects of *Kcnn4* on the changes of intracellular K$^+$ and Ca$^{2+}$ during inflammasome. [K$^+$]$_i$ decrease by LP was abrogated by *Kcnn4* KO (*Figure 6F*, *Figure 6—source data 1*), consistent with the roles of KCa3.1 channel in K$^+$ efflux. Further, [Ca$^{2+}$]$_{Lys}$ recovery by LP removal was inhibited by TRAM-34 (*Figure 6G*, *Figure 6—source data 1*), indicating that K$^+$ efflux contributes to [Ca$^{2+}$]$_{Lys}$ recovery, likely through facilitation of SOCE and subsequent ER→lysosome Ca$^{2+}$ refilling. We also studied physical coupling between Ca$^{2+}$ influx and K$^+$ efflux channels, in addition to their functional coupling. PLA demonstrated physical association between KCNN4 and ORAI1 channel by LP (*Figure 6—figure supplement 1D*, *Figure 6—figure supplement 1—source data 1*), which might facilitate functional coupling between K$^+$ efflux and SOCE (*Ferreira and Schlichter, 2013*). Coupling between KCNN4 and ORAI1 was abrogated by BAPTA-AM (*Figure 6—figure supplement 1D*, *Figure 6—figure supplement 1—source data 1*), suggesting the roles of increased Ca$^{2+}$ in their directional movement and contact.

We also studied the roles of KCa3.1 channel in inflammasome by other stimulators. Inflammasome by LLOMe or MSU, lysosomotropic agents, was significantly inhibited by *Kcnn4* KO; however, that by nigericin directly promoting K$^+$ efflux or ATP inducing K$^+$ efflux through pannexin-1 channel (*Xu et al., 2020*; *Yang et al., 2015*) was not significantly affected (*Figure 6—figure supplement 1E*, *Figure 6—figure supplement 1—source data 1*), suggesting that KCa3.1 channel is crucial in K$^+$ efflux associated with lysosomotropic agents or lysosomal Ca$^{2+}$ channels. Since K$^+$ efflux is crucial in NLRP3 binding to NEK7 and oligomerization (*He et al., 2016*), we studied the roles of KCa3.1 channel in NLRP3 oligomerization. NLPR3 oligomerization by LP was abrogated by *Kcnn4* KO (*Figure 6H*, *Figure 6—source data 2*). NLRP3 binding to NEK7 (*He et al., 2016*), which was observed in control MΦs treated with LP by immunoprecipitation, was also abrogated by *Kcnn4* KO or TRAM-34 (*Figure 6—figure supplement 1F and G*, *Figure 6—figure supplement 1—source data 2*), suggesting the roles of KCa3.1 channel in NLRP3 interaction with NEK7 and oligomerization.

We next studied whether *Kcnn4*-KO mice are resistant to HFD-induced metabolic inflammation. Consistent with the roles of KCa3.1 channel in inflammasome by LP, *Kcnn4*-KO mice on HFD showed significantly improved glucose tolerance (*Figure 6—figure supplement 1H*, *Figure 6—figure supplement 1—source data 1*). Inflammasome and metabolic inflammation by HFD were also significantly ameliorated by *Kcnn4* KO as evidenced by attenuated caspase-1 cleavage and significantly reduced the number of ASC specks or CLS in WAT (*Figure 6I–K*, *Figure 6—source data 1 and 2*).

## Mechanism of Ca$^{2+}$-induced inflammasome activation

We next investigated the mechanism of inflammasome by increased [Ca$^{2+}$]$_i$. We studied whether TAK1-JNK activation observed in inflammasome by lysosomal rupture releasing lysosomal Ca$^{2+}$ (*Okada et al., 2014*) participates in inflammasome by LP. JNK was activated by LP treatment for 21 hr (*Figure 7A*, *Figure 7—source data 2*). Since JNK activation occurs after LPS treatment alone for a

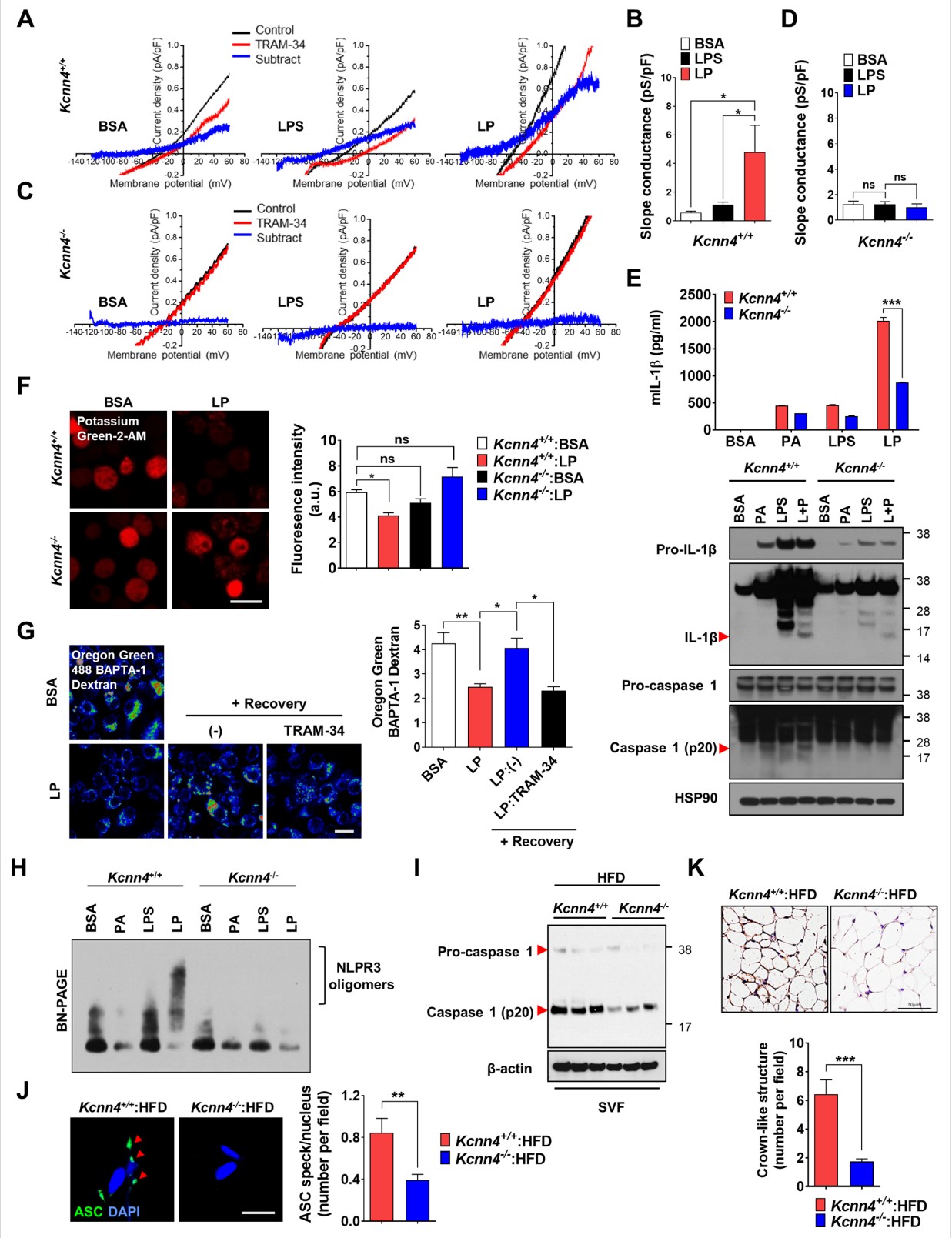

**Figure 6.** Role of KCa3.1 Ca²⁺-activated K⁺ channel in inflammasome. (**A–D**) Nystatin-perforated patch clamp and slope conductance of TRAM-34-sensitive I/V curve in *Kcnn4*+/+ (**A, B**) or *Kcnn4*-/- MΦs (**C, D**) treated with lipopolysaccharide (LPS) alone for 4 hr or with LP for a total of 4 hr including LPS pretreatment for 3 hr (**B, D**). Representative I/V curves (**A, C**). (n = 13 for each *Kcnn4*+/+ group; n = 4 for *Kcnn4*-/-:BSA; n = 5 for *Kcnn4*-/-:LPS; n = 4 for *Kcnn4*-/-:LP). (**E**) IL-1β ELISA of culture supernatant (upper) and immunoblotting (IB) of cell lysate using indicated Abs (lower) after treating *Kcnn4*+/+

*Figure 6 continued on next page*

*Figure 6 continued*

or *Kcnn4$^{-/-}$* MΦs with palmitic acid (PA) alone or LPS alone for 21 hr, or LP for a total of 21 hr including LPS pretreatment for 3 hr (n = 3). (**F**) [K$^+$]$_i$ in *Kcnn4$^{+/+}$* or *Kcnn4$^{-/-}$* MΦs treated with LP for a total of 21 hr including LPS pretreatment for 3 hr (right). Representative Potassium Green-2 images (left) (n = 5). (**G**) Oregon Green BAPTA-1 Dextran (OGBD)-loaded bone marrow-derived macrophages (BMDMs) were treated with LP for a total of 4 hr including LPS pretreatment for 3 hr. [Ca$^{2+}$]$_{Lys}$ recovery after LP removal with or without TRAM-34 (right). Representative fluorescence images (left) (n = 11 for BSA; n = 8 for LP; n = 6 for LP:(-); n = 9 for LP:TRAM-34). (**H**) BN gel electrophoresis and subsequent IB of lysate of MΦs treated with LPS alone for 21 hr or with LP for a total of 21 hr including LPS pretreatment for 3 hr using indicated Ab. (**I**) IB of stromal vascular fraction (SVF) of while adipose tissue (WAT) from mice fed high-fat diet (HFD) for 8 weeks using indicated Abs (n = 3). (**J**) The number of ASC specks in WAT of mice of (**I**) identified by ASC immunofluorescence (right). Representative ASC specks (red arrow heads) (left) (scale bar, 20 μm) (n = 28 for *Kcnn4$^{+/+}$*:HFD; n = 22 for *Kcnn4$^{-/-}$*:HFD). (**K**) The number of crown-like structures (CLS) in WAT of mice of (**I**) identified by F4/80 immunohistochemistry (lower). Representative F4/80 immunohistochemistry (upper) (scale bar, 50 μm) (n = 12 for *Kcnn4$^{+/+}$*:HFD; n = 11 for *Kcnn4$^{-/-}$*:HFD). Data shown as means ± SEM from more than three independent experiments. *p<0.05, **p<0.01, and ***p<0.001 by one-way ANOVA with Tukey's test (**B, D, F, G**), or two-way ANOVA with Sidak test (**E**). Scale bar, 20 μm.

The online version of this article includes the following source data and figure supplement(s) for figure 6:

**Source data 1.** Original data for slope conductance, fluorescence, and the numbers of ASC specks and CLS.

**Source data 2.** Original uncropped blot.

**Figure supplement 1.** Effects of *Kcnn4* KO on metabolic profile and inflammasome.

**Figure supplement 1—source data 1.** Original data for fold change of mRNA expression, ELISA, the number of PLA dots, and blood glucose.

**Figure supplement 1—source data 2.** Original uncropped blot.

short time and then wanes (*An et al., 2017*), we studied the time sequence of JNK activation by LP. JNK activation by LPS alone occurred 30 min after treatment, subsided since 2 hr, and never occurred again (*Figure 7A*, *Figure 7—source data 2*). In contrast, JNK activation occurred again after LP treatment for 21 hr (*Figure 7A*, *Figure 7—source data 2*), suggesting different mechanism and time scale of JNK activation depending on additional events such as lysosomal ones. While we determined [Ca$^{2+}$]$_i$ after LP treatment for a total 4 hr including LPS pretreatment for 3 hr (actual LP treatment time is 1 hr) throughout the study as Ca$^{2+}$ flux is expected to occur since early time, [Ca$^{2+}$]$_i$ further increased 21 hr after LP treatment (*Figure 7—figure supplement 1A*, *Figure 7—figure supplement 1—source data 1*), which could be high enough to act as signal 2 of inflammasome. [K$^+$]$_i$ decrease was also observed not 4 but 21 hr after LP treatment (*Figure 7—figure supplement 1A*, *Figure 7—figure supplement 1—source data 1*), suggesting that Ca$^{2+}$ influx supported by K$^+$ efflux is pronounced at later time. Additionally, [Ca$^{2+}$]$_{ER}$ was reduced not 4 but 21 hr after LP treatment without extracellular Ca$^{2+}$ removal (*Figure 7—figure supplement 1A*, *Figure 7—figure supplement 1—source data 1*), suggesting full alteration of intracellular Ca$^{2+}$ distribution at later time of LP treatment. Since JNK contributes to inflammasome through ASC phosphorylation (*Hara et al., 2013*), we studied the relationship between JNK and ASC phosphorylation/oligomerization. Intriguingly, ASC phosphorylation and oligomerization occurred 21 hr after LP treatment but not 30 min after LPS treatment despite similar JNK activation (*Figure 7B and C*, *Figure 7—source data 2*), suggesting that JNK activation 21 hr after LP treatment leads to ASC phosphorylation and oligomerization likely due to signal 2 such as lysosomal events. Indeed, JNK activation by LP treatment for 21 hr was markedly suppressed by *Kcnn4* or *Trpm2* KO (*Figure 7—figure supplement 1B*, *Figure 7—figure supplement 1—source data 2*), supporting that events such as lysosomal Ca$^{2+}$ release and/or K$^+$ efflux are necessary for delayed JNK activation. SP600125, a JNK inhibitor, suppressed ASC phosphorylation and oligomerization by LP (*Figure 7D*, *Figure 7—source data 2*, *Figure 7—figure supplement 1C*, *Figure 7—figure supplement 1—source data 2*), indicating that JNK activation is necessary but not sufficient for ASC phosphorylation/oligomerization, as shown by no ASC phosphorylation/oligomerization 0.5 hr after LPS treatment despite strong JNK activation (*Figure 7B and C*, *Figure 7—source data 2*). Consistent with the roles of JNK in inflammasome activation by LP, IL-1β release by LP was inhibited by JNK inhibitor (*Figure 7E*, *Figure 7—source data 1*).

When we studied the roles of TAK1 as JNK upstream (*Okada et al., 2014*), IL-1β release by LP was not inhibited by 5Z-7-oxozeaenol, a TAK1 inhibitor (*Figure 7—figure supplement 1D*, *Figure 7—figure supplement 1—source data 1*), in contrast to inflammasome activation by lysosomal rupture (*Okada et al., 2014*). We thus studied the roles of ASK1 participating in inflammasome activation by diverse inflammasome activators such as bacteria or virus (*Immanuel et al., 2019*; *Place et al., 2018*). IL-1β release by LP was significantly inhibited by selonsertib, an ASK1 inhibitor (*Figure 7E*, *Figure*

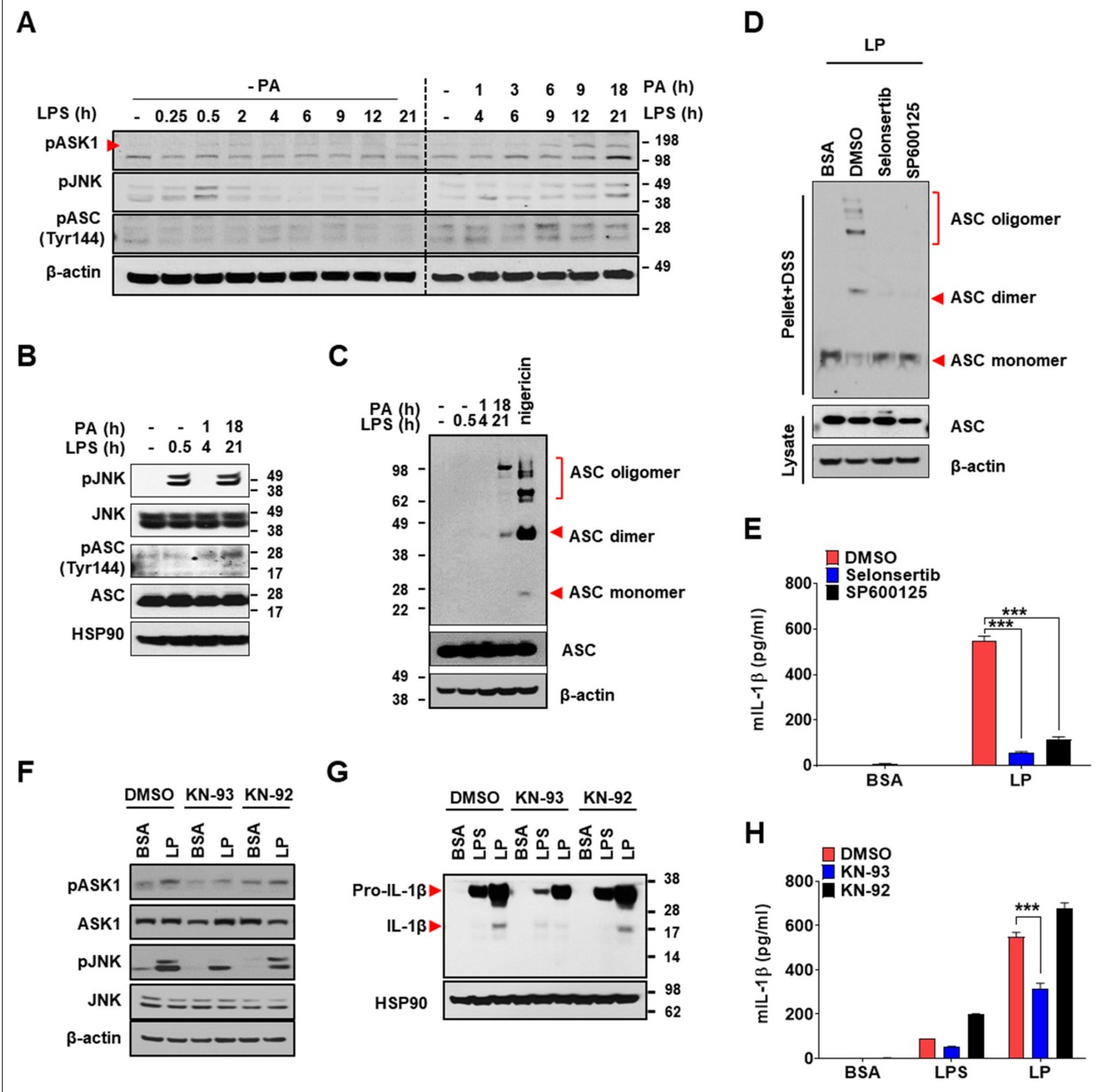

**Figure 7.** Mechanism of Ca$^{2+}$-mediated inflammasome. (**A**) Immunoblotting (IB) of bone marrow-derived macrophages (BMDMs) treated with lipopolysaccharide (LPS) alone without palmitic acid (PA) ('- PA') for indicated time period (left half) or with 'PA' (together with LPS) for indicated time period after LPS pretreatment for 3 hr (right half) (hence, the numbers indicating LPS treatment time in the right half are 3+PA treatment time), using indicated Abs. (**B**) IB of BMDMs treated with LPS alone for 0.5 hr or 'PA' (together with LPS) for 1 or 18 hr after LPS pretreatment for 3 hr (hence, the numbers indicating LPS treatment time of 4 or 21 hr are 3+PA treatment time), using indicated Abs. (**C**) BMDMs were treated with LPS alone for 0.5 hr, 'PA' (together with LPS) for 1 or 18 hr after LPS pretreatment for 3 hr (hence, the numbers indicating LPS treatment time of 4 or 21 hr are 3+PA treatment time) or nigericin for 45 min after LPS pretreatment for 3 hr. IB using indicated Abs after disuccinimidyl suberate (DSS) crosslinking. (**D**) BMDMs were treated with LP for a total of 21 hr including LPS pretreatment for 3 hr in the presence or absence of ASK1 (selonsertib) or JNK inhibitor (SP600125). IB using indicated Abs after DSS crosslinking. (**E**) IL-1β ELISA of culture supernatant after treating BMDMs with LP for a total of 21 hr including LPS pretreatment for 3 hr in the presence or absence of selonsertib or SP600125. (**F–H**) IB using indicated Abs (**F, G**) and IL-1β ELISA of culture supernatant (**H**) after treating BMDMs with LPS alone for 21 hr or with LP for a total of 21 hr including LPS pretreatment for 3 hr in the presence or absence of KN-93 or -92 (n = 3). Data shown as means ± SEM from more than three independent experiments. ***p<0.001 by two-way ANOVA with Tukey's test (**E, H**).

The online version of this article includes the following source data and figure supplement(s) for figure 7:

**Source data 1.** Original data for ELISA.

**Source data 2.** Original uncropped blot.

*Figure 7 continued on next page*

*Figure 7 continued*
**Figure supplement 1.** Effect of *Kcnn4* KO and inhibitor of TAK1 or ASK1 on activation of inflammasome and JNK.
**Figure supplement 1—source data 1.** Original data for fluorescence (ratio) and ELISA.
**Figure supplement 1—source data 2.** Original data blot.

*7—source data 1*), suggesting the roles of ASK1 in inflammasome activation by LP. Consistently, JNK activation by LP was inhibited by selonsertib (*Figure 7—figure supplement 1E*, *Figure 7—figure supplement 1—source data 2*). Further, ASK1 activation was observed at later time of LP treatment (*Figure 7A*, *Figure 7—source data 2*), suggesting that LP induces ASK1 activation likely through additional mechanisms such as lysosomal events at JNK upstream. Consistent with the roles of ASK1 in JNK activation, ASK1 inhibitor blocked ASC phosphorylation/oligomerization as efficiently as JNK inhibitor (*Figure 7D*, *Figure 7—source data 2*, *Figure 7—figure supplement 1C*), supporting that activated ASK1 induces JNK activation and subsequent ASC phosphorylation/oligomerization.

When we studied the mechanism of ASK1 activation by LP, ASK1 activation by LP was attenuated by KN-93, a CaMKII inhibitor, but not by KN-92, a KN-93 congener without CaMKII inhibitory activity (*Figure 7F*, *Figure 7—source data 2*), which supports the roles of increased $[Ca^{2+}]_i$ and subsequent CAMKII in ASK1 activation by LP, similar to the inhibition of lysosomal rupture-induced inflammasome by KN-93 (*Okada et al., 2014*). JNK phosphorylation and inflammasome activation as evidenced by maturation or release of IL-1β by LP were also inhibited by KN-93 but not by KN-92 (*Figure 7F–H*, *Figure 7—source data 1 and 2*). These results indicate that increased $[Ca^{2+}]_i$ due to lysosomal $Ca^{2+}$

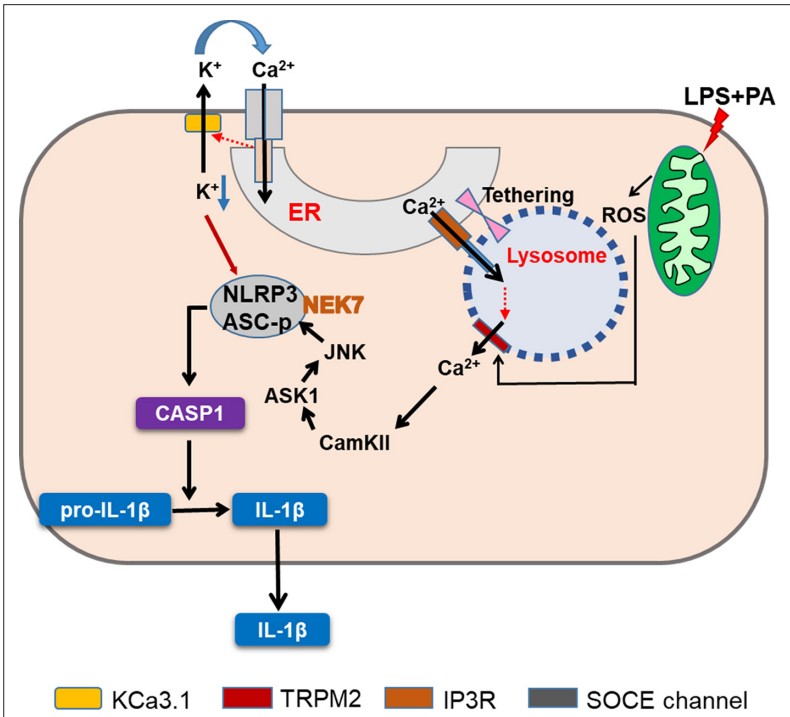

**Figure 8.** Graphic summary. LP, an effector combination activating inflammasome related to metabolic inflammation, induces generation of mitochondrial reactive oxygen species (ROS), which activates TRPM2 channel on lysosome and releases lysosomal $Ca^{2+}$. ER→lysosome $Ca^{2+}$ refilling facilitated by ER-lysosome tethering replenishes diminished lysosomal $Ca^{2+}$ content and supports sustained lysosomal $Ca^{2+}$ release. ER emptying due to ER→lysosome $Ca^{2+}$ refilling activates SOCE. SOCE, in turn, is positively modulated by $K^+$ efflux through KCa3.1, a $Ca^{2+}$-activated $K^+$ efflux channel, mediating hyperpolarization-induced acceleration of extracellular $Ca^{2+}$ influx. $Ca^{2+}$ release from lysosome activates CaMKII, which induces delayed activation of ASK1 and JNK. Delayed JNK activation leads to ASC phosphorylation and oligomerization, leading to the formation of inflammasome complex together with NLRP3 and NEK7. $K^+$ efflux changes intracellular milieu and induces structural changes of NLRP3 or NLRP3 binding to PI(4)P on dispersed Golgi network, facilitating inflammasome activation. Golgi complex and microtubule-organizing center (MTOC) are not shown for clarity (CASP1, caspase 1).

release facilitated by ER→lysosome $Ca^{2+}$ refilling and $K^+$ efflux through KCa3.1 channel induces delayed activation of ASK1 and JNK, leading to ASC oligomerization or NLRP3 inflammasome by LP and metabolic inflammation.

## Discussion

Agents inducing lysosomal stress or lysosomotropic agents are well-known inflammasome activators (***Hornung et al., 2008***). Regarding the mechanism of inflammasome by lysosomotropic agents, lysosomal $Ca^{2+}$ has been incriminated (***Okada et al., 2014***; ***Weber and Schilling, 2014***), while papers refuting the role of $Ca^{2+}$ have been published (***Katsnelson et al., 2015***). We found that lysosomal $Ca^{2+}$ release and subsequent $[Ca^{2+}]_i$ increase by LP lead to inflammasome through CAMKII-ASK1 and delayed JNK activation, which is critical for metabolic inflammation (***Figure 8***).

Among diverse inflammasome activators, LP is a representative effector combination responsible for metabolic inflammation related to metabolic syndrome. However, the mechanism of inflammasome activation by LP has been unclear. It is well known that PA induces stresses of organelle such as ER and mitochondria (***Bachar et al., 2009***; ***Korge et al., 2003***). We observed that mitochondrial ROS accumulate by LP and activate lysosomal $Ca^{2+}$ release through TRPM2. The roles of mitochondrial ROS in inflammasome activation have been reported ***Weber and Schilling, 2014***; however, causal relationship between mitochondrial ROS and lysosomal events has not been addressed. The roles of TRPM2 in diabetes or β-cell function have been studied (***Uchida and Tominaga, 2011***; ***Zhang et al., 2012***). However, the roles of TRPM2 in metabolic inflammation were not studied.

While we have shown the roles of TRPM2 in inflammasome and metabolic inflammation, TRPM2 exists on both plasma membrane and lysosome (***Wang et al., 2020***). The roles of plasma membrane TRPM2 in inflammasome by high glucose or particulate materials have been reported (***Tseng et al., 2017***; ***Zhong et al., 2013***), while the role of TMRP2 in inflammation activation by lipid stimulators related to metabolic inflammation has not been addressed. However, we observed no activation of TRPM2 current on the plasma membrane of MΦs treated with LP, arguing against the roles of plasma membrane TRPM2 in inflammasome by LP. Further, dissipation of lysosomal $Ca^{2+}$ reservoir by bafilomycin A1 abrogated $[Ca^{2+}]_i$ increase by LP, strongly supporting the roles of lysosomal TRPM2 rather than plasma membrane TRPM2 in inflammation by LP.

We observed decreased $[Ca^{2+}]_{ER}$ in inflammasome activation by LP, which was due to ER→lysosome $Ca^{2+}$ refilling through $IP_3R$ to replenish diminished $[Ca^{2+}]_{Lys}$ after lysosomal $Ca^{2+}$ release. Previous papers have shown the roles of ER $Ca^{2+}$ in inflammasome activation (***Lee et al., 2012***); however, the role of ER $Ca^{2+}$ replenishing reduced $[Ca^{2+}]_{Lys}$ in inflammasome has not been addressed. Direct $Ca^{2+}$ efflux from ER to cytoplasm, which was reported in inflammasome activation by ATP (***Lee et al., 2012***), cannot explain ER→lysosome refilling or abrogation of LP-induced increase of $[Ca^{2+}]_i$ by bafilomycin A1, which would not decrease but increase $[Ca^{2+}]_i$ after ER $Ca^{2+}$ release into cytoplasm due to abrogated lysosomal $Ca^{2+}$ buffering (***López Sanjurjo et al., 2014***). Likely due to ER $Ca^{2+}$ release to replenish reduced $[Ca^{2+}]_{Lys}$, $[Ca^{2+}]_{ER}$ was reduced, which in turn induced SOCE. Then, extracellular $Ca^{2+}$ influx appears to replenish ER $Ca^{2+}$ after ER $Ca^{2+}$ loss due to ER→lysosome $Ca^{2+}$ refilling. The roles of extracellular $Ca^{2+}$ influx in inflammasome have been suggested, while extracellular $Ca^{2+}$ influx was unrelated to ER $Ca^{2+}$ store (***Lee et al., 2012***; ***Tseng et al., 2017***). STIM1 aggregation and its colocalization with ORAI1 strongly support SOCE in inflammasome by LP to replenish ER $Ca^{2+}$ depletion (***Vaca, 2010***), which is different from the activation of $Ca^{2+}$-sensing receptor by extracellular $Ca^{2+}$ reported in ATP-induced inflammasome (***Lee et al., 2012***). We have observed that BTP2 inhibiting $Ca^{2+}$ influx through ORAI1 (***Bogeski et al., 2010***) suppressed LP-induced inflammasome, suggesting that extracellular $Ca^{2+}$ influx is a process of SOCE through STIM1/ORAI1 channel activated by ER $Ca^{2+}$ depletion rather than a direct extracellular $Ca^{2+}$ influx into cytoplasm through TRPM2 on the plasma membrane (***Zhong et al., 2013***). Thus, these results additionally support the roles of lysosomal TRPM2 but not plasma membrane TRPM2 in LP-induced inflammasome.

We observed that $K^+$ efflux occurs in inflammasome by LP, similar to that by other activators. $K^+$ efflux through KCa3.1 channel appears to induce NLRP3 binding to NEK7 and oligomerization, which was inhibited by TRAM-34 or *Kcnn4* KO. Previous papers suggested the roles of KCa3.1 channel in SOCE, while such relationship was unrelated to inflammasome (***Duffy et al., 2015***; ***Gao et al., 2010***). A KCa3.1 channel activator, together with LPS, has been reported to induce IL-1β release; however, the roles of KCa3.1 channel in authentic inflammasome have not been shown (***Schroeder et al.,***

*2017*). Although we observed significant roles of KCa3.1 channel in inflammasome activation, contribution of other $Ca^{2+}$-activated $K^+$ channels such as BK channels cannot be eliminated since iberitoxin, a BK channel inhibitor, has been reported to inhibit ATP-induced inflammasome (*Schroeder et al., 2017*). However, the relationship between BK channel and $Ca^{2+}$ flux was not studied. Previous papers also reported the roles of TWIK2 channel, a two-pore $K^+$ channel, in $K^+$ efflux in ATP-induced inflammasome, which was inhibited by quinine (*Di et al., 2018*). The role of THIK-1, another two-pore $K^+$ channel in IL-1β release from microglia by ATP, has also been reported (*Madry et al., 2018*). However, inflammasome by LP was not inhibited by quinine. Further, the relationship between $K^+$ efflux through two-pore $K^+$ channels and SOCE was not studied. The relationship between ATP-induced P2X7 activation and $K^+$ efflux could be distinct from coupling of $Ca^{2+}$ influx and $K^+$ efflux by lysosomotropic agents or other events primarily affecting lysosomal $Ca^{2+}$ channel (*Di et al., 2018*; *Muñoz-Planillo et al., 2013*). Such differences could explain no effect of quinine on LP-induced inflammasome or undiminished lysosomotropic agents-induced inflammasome in *THIK-1*-KO MΦs or microglia (*Drinkall et al., 2022*).

Altogether, we have shown the sequential events from mitochondrial ROS-induced lysosomal TRPM2 channel activation to ER→lysosome $Ca^{2+}$ refilling and SOCE coupled with KCa3.1 $K^+$ efflux channel activation in inflammasome by LP and metabolic inflammation (*Figure 8*), which could also be applied to that by other lysosomotropic agents. These results suggest mitochondrial ROS-lysosomal TRPM2 axis as a potential therapeutic target for the treatment of metabolic inflammation. Elucidation of KCa3.1 channel as the $K^+$ efflux channel in inflammasome and its role in facilitation of extracellular $Ca^{2+}$ influx through SOCE would provide another target for modulation of inflammasome that is activated in a variety of diseases or conditions in addition to metabolic syndrome. ER→lysosome refilling might have implication in diverse conditions or diseases associated with lysosomal $Ca^{2+}$ changes such as autophagy, vaccination, or inflammation (*Park et al., 2022*; *Tahtinen et al., 2022*).

## Materials and methods

### GCaMP3 $Ca^{2+}$ imaging

BMDMs were kindly provided by Emad S. Alnemri, Thomas Jefferson University, through Je-Wook Yu, Yonsei University (*Fernandes-Alnemri et al., 2009*). Cytokine responses to LPS or LP were periodically monitored to authenticate cell function. Cells were tested negative for mycoplasma contamination. BMDMs grown on four-well chamber were transfected with a plasmid encoding a perilysosomal GCaMP3-ML1 $Ca^{2+}$ probe (*Shen et al., 2012*). Cytokine responses to LPS or LP were periodically monitored. Cells were tested negative for mycoplasma contamination. After 48 hr, cells were treated with LP for 1 hr after LPS pretreatment for 3 hr, and then lysosomal $Ca^{2+}$ release was measured in a basal $Ca^{2+}$ solution containing 145 mM NaCl, 5 mM KCl, 3 mM MgCl$_2$, 10 mM glucose, 1 mM EGTA, and 20 mM HEPES (pH 7.4) by monitoring fluorescence intensity at 470 nm using a LSM780 confocal microscope (Carl Zeiss, LSM 780). GPN was added at the indicated time points.

### Determination of $[Ca^{2+}]_i$

After pretreatment with LPS for 3 hr and treatment with LP for 1 hr, cells were loaded with Fluo-3-AM (Invitrogen) at 37°C for 30 min. $[Ca^{2+}]_i$ was measured in a basal $Ca^{2+}$ solution containing 145 mM NaCl, 5 mM KCl, 3 mM MgCl$_2$, 10 mM glucose, 1 mM EGTA, and 20 mM HEPES (pH 7.4), using a LSM780 confocal microscope (Carl Zeiss).

For ratiometric determination of $[Ca^{2+}]_i$, cells treated with LP were loaded with 2 μM of the acetoxymethyl ester form of Fura-2 (Invitrogen) in RPMI-1640 at 37°C for 30 min and fluorescence data were analyzed using MetaFluor (Molecular Devices) on Axio Observer A1 (Zeiss) equipped with 150 W xenon lamp Polychrome V (Till Photonics), CoolSNAP-Hq2 digital camera (Photometrics), and Fura-2 filter set. Fluorescence at 340/380 nm was measured in phenol red-free RPMI, and converted to $[Ca^{2+}]_i$ using the following equation (*Grynkiewicz et al., 1985*).

$$[Ca^{2+}]_i \ = \ K_d \times [(R - R_{min})/(R_{max} - R)] \times [F_{min(380)}/F_{max(380)}]$$

where $K_d$ = Fura-2 dissociation constant (224 nM at 37°C), $F_{min(380)}$ = 380 nm fluorescence in the absence of $Ca^{2+}$, $F_{max(380)}$ = 380 nm fluorescence with saturating $Ca^{2+}$, R = 340/380 nm fluorescence ratio, $R_{max}$ = 340/380 nm ratio with saturating $Ca^{2+}$, and $R_{min}$ = 340/380 nm ratio in the absence of $Ca^{2+}$.

## Measurement of $[Ca^{2+}]_{Lys}$

To measure $[Ca^{2+}]_{Lys}$, cells were loaded with 100 μg/ml OGBD, an indicator of lysosomal luminal $Ca^{2+}$, at 37°C in the culture medium for 12 hr to allow uptake by endocytosis. After additional incubation for 4 hr without indicator, cells were treated with LP for 1 hr after LPS pretreatment for 3 hr, and then washed in HBS (135 mM NaCl, 5.9 mM KCl, 1.2 mM MgCl₂, 1.5 mM CaCl₂, 11.5 mM glucose, 11.6 mM HEPES, pH 7.3) for confocal microscopy (*Garrity et al., 2016*).

## Measurement of ER $Ca^{2+}$ content ($[Ca^{2+}]_{ER}$)

BMDMs grown on four-well chamber were transfected with GEM-CEPIA1er (Addgene) (*Suzuki et al., 2014*) or a ratiometric FRET-based Cameleon probe D1ER (*Park et al., 2009*) using Lipofectamine 2000. After 48 hr, cells were pretreated with LPS for 3 hr and then treated with LP for 1 hr in a $Ca^{2+}$-free Krebs-Ringer bicarbonate (KRB) buffer (Sigma) to eliminate the effect of extracellular $Ca^{2+}$ influx into ER (*Xu et al., 2015*). Fluorescence was measured using an LSM780 confocal microscope (Zeiss) at an excitation wavelength of 405 nm and an emission wavelength of 466 or 520 nm. F466/F520 was calculated as an index of $[Ca^{2+}]_{ER}$ (*Suzuki et al., 2014*). D1ER fluorescence intensity ratio (F540/F490) was determined using an LSM780 confocal microscope (Zeiss).

## Simultaneous monitoring of $[Ca^{2+}]_{Lys}$ and $[Ca^{2+}]_{ER}$

Twenty-four hours after transfection with GEM-CEPIA1er, BMDMs were loaded with OGBD for 12 hr and chased for 4 hr. After treating cells with LP for a total of 4 hr including LPS pretreatment for 3 hr, medium was changed to a fresh one without LP. Cells were then monitored for $[Ca^{2+}]_{Lys}$ and $[Ca^{2+}]_{ER}$ in a $Ca^{2+}$-free KRB buffer using an LSM780 confocal microscope.

## Proximity ligation assay

Contact between ER and lysosome was examined using Duolink In Situ Detection Reagents Red kit (Sigma) according to the manufacturer's protocol. Briefly, BMDMs treated with test agents were incubated with antibodies (Abs) to ORP1L (Abcam, 1:200) and VAP-A (Santa Cruz, 1:200), or with those to KCNN4 (Invitrogen) and ORAI1 (Novusbio) at 4°C overnight. After washing, cells were incubated with PLA plus and minus probes at 37°C for 1 hr. After ligation reaction to close the circle and rolling circle amplification (RCA) of the ligation product, fluorescence-labeled oligonucleotide hybridized to RCT product was observed by fluorescence microscopy.

## SOCE channel activation

BMDMs transfected with YFP-STIM1 and 3xFLAG-mCherry Red-Orai1/P3XFLAG7.1 (kindly provided by Joseph Yuan, University of North Texas Health Science Center, USA, through Cha S-G, Yonsei University) were treated with LP for a total of 4 hr including LPS pretreatment for 3 hr, which were then subjected to confocal microscopy to visualize STIM1 puncta and their co-localization with ORAI1.

## Abs and IB

Cells or tissues were solubilized in a lysis buffer containing protease inhibitors. Protein concentration was determined using Bradford method. Samples (10–30 μg) were separated on 4–12% Bis-Tris gel (NUPAGE, Invitrogen), and transferred to nitrocellulose membranes for IB using the ECL method (Dongin LS). For IB, Abs against the following proteins were used: IL-1β (R&D Systems, AF-401-NA, 1:1000), caspase 1 p20 (Millipore, ABE1971, 1:1000), ASC (Adipogen AL177, 1:1000), phospho-JNK (Cell Signaling #9251, 1:1000), JNK (Santa Cruz sc7345, 1:1000), phospho-ASK1 (Invitrogen PA5-64541, 1:1000), ASK1 (Abcam ab45178, 1:1000), phospho-ASC (ECM Biosciences AP5631, 1:1000), NLRP3 (Invitrogen MA5-23919, 1:1000), NEK7 (Abcam ab133514, 1:1000), HSP 90 (Santa Cruz sc13119, 1:1000), and β-actin (Santa Cruz sc47778, 1:1,000).

## Immunoprecipitation

After lysis of cells in an ice-cold lysis buffer (400 mM NaCl, 25 mM Tris-HCl, pH 7.4, 1 mM EDTA, and 1% Triton X-100) containing protease and phosphatase inhibitors, lysates were centrifuged at 12,000

× *g* for 10 min in microfuge tubes, and supernatant was incubated with anti-NEK7 (Abcam, 1:1000) Ab or control IgG in binding buffer (200 mM NaCl, 25 mM Tris-HCl, pH 7.4, 1 mM EDTA) with constant rotation at 4°C for 1 hr. After adding 50 µl of 50% of Protein-G bead (Roche Applied Science) to lysates and incubation with rotation at 4°C overnight, resins were washed with binding buffer. After resuspending pellet in a sample buffer (Life Technology) and heating at 100°C for 3 min, supernatant was collected by centrifugation at 12,000 × *g* for 30 s, followed by electrophoretic separation in a NuPAGE gradient gel (Life Technology). IB was conducted by sequential incubation with anti-NEK7 or -NLRP3 Ab as the primary Ab and horseradish peroxidase-conjugated anti-rabbit IgG or -mouse IgG. Bands were visualized using an ECL kit.

## Detection of ASC oligomerization

BMDMs were washed in ice-cold PBS, and then lysed in NP-40 buffer (20 mM HEPES-KOH pH 7.5, 150 mM KCl, 1% NP-40, and protease inhibitors). Lysate was centrifuged at 2000 × *g*, 4°C for 10 min. Pellets were washed and resuspended in PBS containing 2 mM disuccinimidyl suberate (DSS) for crosslinking, followed by incubation at room temperature for 30 min. Samples were then centrifuged at 2000 × *g*, 4°C for 10 min. Precipitated pellets and soluble lysates were subjected to IB using anti-ASC Ab.

## Blue Native PAGE

Blue Native polyacrylamide gel electrophoresis (BN-PAGE) was performed using Bis-Tris Native-PAGE system (Invitrogen, Carlsbad, CA), according to the manufacturer's instructions. Briefly, cells were collected and lysed in 1× NativePAGE Sample Buffer (Invitrogen) containing 1% digitonin and protease inhibitor cocktail, followed by centrifugation at 13,000 rpm, 4°C for 20 min. 20 µl supernatant mixed with 1 µl 5% G-250 Sample Additive was loaded on a NativePAGE 3~12% Bis-Tris gel. Samples separated on gels were transferred to PVDF membranes (Millipore, Darmstadt, Germany) using transfer buffer, followed by IB using anti-NLRP3 Ab.

## Immunofluorescence study

Cells were grown on four-chamber plates. After treatments, cells were fixed with 4% paraformaldehyde for 15 min and permeabilized with 0.5% triton X-100 for 15 min. After blocking with 5% goat serum for 1 hr, cells were incubated with anti-ASC at 4°C overnight. On the next day, samples were incubated with Alexa 488-conjugated anti-mouse or anti-rabbit IgG Ab (Invitrogen) for 1 hr. After nuclear staining with DAPI (Invitrogen), cells were subjected to confocal microscopy (Carl Zeiss, LSM 780).

## Measurement of $[K^+]_i$

BMDMs treated with LP for a total of 21 hr including LPS pretreatment for 3 hr were labeled with 5 µM Asante Potassium Green-2-AM (Abcam) at 37°C for 30 min. After washing twice with PBS, cells were subjected to confocal microscopy (Carl Zeiss, LSM 780).

## ELISA of cytokines

Cytokine content in culture supernatants of BMDMs or peritoneal MΦs was determined using mouse ELISA kits (R&D Systems), according to the manufacturer's instruction.

## RNA extraction and real-time RT-PCR

Total RNA was extracted from cells or tissues using TRIzol (Invitrogen), and cDNA was synthesized using MMLV Reverse Transcriptase (Promega), according to the manufacturer's protocol. Real-time RT-PCR was performed using SYBR green (Takara) in QuantStudio3 Real-Time PCR System (Applied Biosystems). All expression values were normalized to *GAPDH* or *Rpl32* mRNA level.

Primer sequences are as follows: *Kcnn4*-F, 5'-AACTGGCATCGGACTCATGGTT-3'; *Kcnn4*-R, 5'-AGTCATGAACAGCTGGACCTC-3'.

## Animals

Eight-week-old male *Trpm2*-KO mice (kindly provided by Yasuo Mori, Kyoto University, Japan) were maintained in a 12 hr light/12 hr dark cycle and fed HFD for 12 weeks. During the observation period,

mice were monitored for glucose profile and weighed. *Kcnn4*-KO mice were from Jackson Laboratories. All animal experiments were conducted in accordance with the Public Health Service Policy in Humane Care and Use of Laboratory Animals. Mouse experiments were approved by the IACUC of the Department of Laboratory Animal Resources of Yonsei University College of Medicine, an AAALAC-accredited unit (IACUC No 2018-0083).

## Cell culture and drug treatment

BMDMs were cultured in DMEM supplemented with 10% fetal bovine serum (FBS), 100 U/ml penicillin and 100 μg/ml streptomycin (Lonza). For drug treatment, the following concentrations were used: 50 or 100 ng/ml LPS (Sigma), 300 μM PA (Sigma), 50 μM BAPTA-AM (Invitrogen), 10 μM nigericin (Sigma), 3 mM ATP (Roche), 25 μg/ml MSU (InvivoGen), 0.3 mM LLOMe (Sigma), 3 μM Xestospongin C (Abcam), 10 μM dantrolene (Sigma), 10 μM TPEN (Sigma), 3 mM EGTA (Sigma), 100 nM 2-APB (Sigma), 10 μM BTP2 (Merk Millipore), 5 mM NAC (Sigma), 100 μM MitoTEMPOL (Sigma), 10 μM selonsertib (Selleckchem), 10 μM SP600125 (Sigma), 100–500 nM 5Z-7-oxozeaenol (Sigma), 10 μM KN-93 (Tocris), 10 μM KN-92 (Tocris), 30 μM apigenin (Sigma), quercetin (Sigma), 1 μM bafilomycin A1, 10 μM TRAM-34 (Sigma), 10 μM ACA (Sigma), 100 nM apamin (Sigma), 1 μM paxillin (Sigma), 100 μM quinine (Sigma), 1 mM barium sulfate (Sigma), 10 nM UCL 1684 (Sigma), and 10 μM 4-AP (Sigma). PA stock solution (50 mM) was prepared by dissolving in 70% ethanol and heating at 55°C. Working solution was made by diluting PA stock solution in 2% fatty acid-free BSA-DMEM or -RPMI.

## Peritoneal MΦs

Mice of C57BL/6 background were injected intraperitoneally with 3 ml of 3.85% Brewer thioglycollate. Three days after injection, peritoneal MΦs were harvested, seeded at 6-well plates and maintained in RPMI containing 10% FBS, 100 U/ml penicillin, and 100 μg/ml streptomycin at 37°C for 24 hr in a humid atmosphere of 5% $CO_2$ before treatment. Peritoneal MΦs were used as the primary MΦs throughout the study.

## Stromal vascular fraction (SVF)

SVF of epididymal adipose tissue was prepared as described (*Lee et al., 2016*). Briefly, after cutting SVF into small pieces and incubation in a 2 mg/ml collagenase D solution (Roche) at 37°C in a water bath for 45 min, digested tissue was centrifuged at 1000 × *g* for 8 min. After filtering through a 70 μm mesh and lysis of RBC, SVF was suspended in PBS supplemented with 2% BSA (Sigma), 2 mM EDTA (Cellgro) for further experiments.

## Metabolic studies

IPGTT was performed by intraperitoneal injection of 1 g/kg glucose solution after overnight fasting. Blood glucose concentrations were determined using One Touch glucometer (LifeScan) before (0 min) and 15, 30, 60, 120, and 180 min after glucose injection. Serum insulin was measured using Mouse Insulin ELISA kit (TMB) (AKRIN-011T, Shibayagi, Gunma, Japan). HOMA-IR index was calculated according to the following formula: [fasting insulin (μIU/ml)×fasting glucose (mg/dl)]/405.

## Histology and immunohistochemistry

Tissue samples were fixed with 10% buffered formalin and embedded in paraffin. Sections of 5 μm thickness were stained with H&E for morphometry, or immunostained with F4/80 Ab (Abcam) to detect MΦ aggregates surrounding adipocytes (crown-like structures, CLS).

## Intracellular ROS

To determine ROS, cells were treated with LPS alone for 21 hr, PA alone for 21 hr, or LP for a total of 21 hr including LPS pretreatment for 3 hr in the presence or absence of 5 mM NAC. After incubation with 5 μM CM-H2DCFDA (Invitrogen) at 37°C for 30 min in culture media without FBS and recovery in a complete media for 10 min, confocal microscopy was conducted. To study mitochondria-specific ROS, cells were treated with LPS alone for 21 hr, PA alone for 21 hr, or LP for a total of 21 hr including LPS pretreatment for 3 hr in the presence or absence of 100 μM MitoTEMPOL. After staining with 5 μM MitoSOX (Invitrogen) at 37°C for 30 min and suspension in PBS-1% FBS, cells were subjected

to flow cytometry on FACSVerse (BD Biosciences), and data were analyzed using FlowJo software (TreeStar).

## Measurement of KCa3.1 and TRPM2 current

To measure KCa3.1 channel activity with intact cytosolic $Ca^{2+}$ environment, nystatin perforated whole-cell patch was performed. 180–250 µg/ml of nystatin was added to intracellular pipette solution (140 mM KCl, 5 mM NaCl, 0.5 mM $MgCl_2$, 3 mM MgATP, and 10 mM HEPES, pH adjusted to 7.3 with KOH). The extracellular bath solution (145 mM NaCl, 3.6 mM KCl, 1.3 mM $CaCl_2$, 1 mM $MgCl_2$, 5 mM glucose, 10 mM HEPES, and 10 mM sucrose, pH adjusted to 7.3 with NaOH) was perfused through recording chamber. After perforated whole-cell configuration is formed, KCa3.1 current was activated by applying ramp-pulse voltage from –120 to 60 mV. Voltage-dependent $K^+$ channels were suppressed by –10 mV holding potential. Intrinsic KCa3.1 current activity was isolated by application of a selective inhibitor, TRAM-34. Activity of TRAM-34-sensitive current was analyzed with slope conductance fitting between –60 to –40 mV.

To measure plasma TRPM2 channel activity, conventional whole-cell patch was performed. To induce TRPM2 current, 200 µM of ADP-ribose was added to intracellular pipette solution (87 mM Cs-glutamate, 38 mM CsCl, 10 mM NaCl, 10 mM HEPES, 1 mM EGTA, and 0.9 mM $CaCl_2$, pH adjusted to 7.2 with CsOH). Cells were perfused with extracellular bath solution (143 mM NaCl, 5.4 mM KCl, 0.5 mM $MgCl_2$, 1.8 mM $CaCl_2$, 10 mM HEPES, 0.5 mM $NaH_2PO_4$, and 10 mM glucose, pH 7.4 adjusted with NaOH) during TRPM2 current recording. When maximal TRPM2 current was established, 20 µM ACA was added to inhibit TRPM2 current.

Patch clamp experiments were performed at room temperature. Patch clamp pipettes were pulled to resistances of 2–3 MΩ with PP-830 puller (Narishige, Japan). Electrophysiological data was recorded and analyzed with Axopatch 200B, Digidata 1440A, and Clampfit 11 program (Axon Instruments, CA).

## Statistical analysis

All values are expressed as the means ± s.e.m. from ≥3 independent experiments performed in triplicate. Statistical significance was tested with two-tailed Student's $t$-test to compare values between two groups. One-way ANOVA with Tukey's test was employed to compare between multiple groups. Two-way ANOVA with Bonferroni, Sidak, or Turkey's test were employed to compare multiple repeated measurements between groups. All analyses were performed using GraphPad Prism Version 8 software (La Jolla, CA). p values <0.05 were considered to represent statistically significant differences.

## Acknowledgements

This study was supported by the Basic Science Research Program through the National Research Foundation of Korea funded by the Ministry of Education (NRF-2019R1I1A1A01063850 to HK), and National Research Foundation of Korea (NRF) grant funded by the Korea government (MSIT) (NRF-2019R1A2C3002924 and Rc-2023-00219563 to M-SL). M-SL is the recipient of National Research Foundation of Korea (NRF) grant funded by the Korea government (MSIT) (Rs-2024-00336581).

## Additional information

### Competing interests

Myung-Shik Lee: CEO of LysoTech, Inc. The other authors declare that no competing interests exist.

### Funding

| Funder | Grant reference number | Author |
|---|---|---|
| National Research Foundation of Korea | NRF-2019R1A2C3002924 | Myung-Shik Lee |
| National Research Foundation of Korea | Rs-2023-00219563 | Myung-Shik Lee |

| Funder | Grant reference number | Author |
|---|---|---|
| National Research Foundation of Korea | Rs-2024-00336581 | Myung-Shik Lee |
| National Research Foundation of Korea | 2019R1I1A1A01063850 | Hyereen Kang |

The funders had no role in study design, data collection and interpretation, or the decision to submit the work for publication.

## Author contributions

Hyereen Kang, Seong Woo Choi, Soo-Jin Oh, Investigation; Joo Young Kim, Supervision, Investigation; Sung Joon Kim, Conceptualization, Supervision; Myung-Shik Lee, Conceptualization, Supervision, Project administration

## Author ORCIDs

Hyereen Kang ● https://orcid.org/0000-0001-9412-2145
Soo-Jin Oh ● http://orcid.org/0000-0001-9267-2039
Myung-Shik Lee ● http://orcid.org/0000-0003-3292-1720

## Ethics

All animal experiments were conducted in accordance with the Public Health Service Policy in Humane Care and Use of Laboratory Animals. Mouse experiments were approved by the IACUC of the Department of Laboratory Animal Resources of Yonsei University College of Medicine, an AAALAC-accredited unit (IACUC No 2018-0083).

Reviewer #1 (Public review): https://doi.org/10.7554/eLife.87561.3.sa1
Reviewer #2 (Public review): https://doi.org/10.7554/eLife.87561.3.sa2
Author response https://doi.org/10.7554/eLife.87561.3.sa3

# Additional files

## Supplementary files

• MDAR checklist

## Data availability

All data generated or analyzed during this study are included in the manuscript, supplementary data, and source data for Figure 1–7 and their accompanying figure supplements.

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
