## [Editor Report · eLife assessment]

This **useful** study proposes a role of lysosomal Ca^2+^ release in inflammasome signaling and metabolic inflammation. While the proposed model would be of considerable interest to the field of immunology if validated, the experimental approaches to study calcium dynamics are problematic, with one of several concerns being the transfection efficiency. The major claims of the article are thus only **incompletely** supported.

---

## [Referee Report · Reviewer #1 (Public review)]

This manuscript proposes a complex incoherent model involving Ca2+ signaling in inflammasome activation. The experimental approaches used to study the calcium dynamics are highly problematic and the results shown are of very poor quality.

Major concerns:

(1) The analysis of lysosomal Ca2+release is being carried out after many hours of treatment. Such evidence is not meaningful to claim that PA activates Ca2+ efflux from lysosome and even if this phenomenon was robust, it is not doubtful that such kinetics are meaningful for the regulation of inflammasome activation. Furthermore, the evidence for lysosomal Ca2+ release is indirect and relies on a convoluted process that doesn't make any conceptual sense to me. In addition to these major shortcomings, the indirect evidence of perilysosomal Ca2+ elevation is also of very poor quality and from the standpoint of my expertise in calcium signaling, the data are incredulous. The use of GCaMP3-ML1, *transiently transfected* into BMDMs is highly problematic. The efficiency of transfection in BMDMs is always extremely low and overexpression of the sensor in a few rare cells can lead to erroneous observations. The overexpression also results in gross mislocalization of such membrane-bound sensors. The accumulation of GCaMP3-ML1 in the ER of these cells would prevent any credible measurements of perilysosomal Ca2+ signals. A meaningful investigation of this process in primary macrophages requires the generation of a mouse line wherein the sensor is expressed at low levels in myeloid cells, and shown to be localized almost exclusively in the lysosomal membrane. The mechanistic framework built around these major conceptual and technical flaws is not especially meaningful and since these are foundational results, I cannot take the main claims of this study seriously.

A few transfected cells may overexpress the protein through a strong promoter but this is not ideal. For reliable Ca2+ measurements, one needs low expression of the sensor in a substantially high percentage of cells. This can only be demonstrated by showing the time lapse of Ca2+ responses in the macrophages. More generally, I have nearly 2 decades of experience working with primary BMDMs and it is widely known that primary BMDMs are incredibly difficult to transfect - it is the nature of these cells. The claim that they get high efficiency of transfection is frankly too incredulous to take seriously.

(2) The cytosolic Ca2+ imaging shown in figure 1C doesn't make any sense. It looks like a snapshot of basal Ca2+ many hours after PA treatment - calcium elevations are highly dynamic. Snapshot measurements are not helpful and analyses of Calcium dynamics requires a recording over a certain timespan. Unfortunately, this technical approach has been used throughout the manuscript. Also, BAPTA-AM abrogates IL-1b secretion because IL-1b transcription is Ca2+ dependent - the result shown in figure 1D does not shed light on anything to do with inflammasome activation and it is misleading to suggest that.

(3) Trpm2-/- macrophages are known to be hyporesponsive to inflammatory stimuli - the reduced secretion of IL-1b by these macrophages is not novel. From a mechanistic perspective, this study does not add much to that observation and the proposed role of TRPM2 as a lysosomal Ca2+ release channel is not substantiated by good quality Ca2+ imaging data (see point 3 above). Furthermore, the study assumes that TRPM2 is a lysosomal ion channel. One paper reported TRPM2 in the lysosomes but this is a controversial claim, with no replication or further development in the last 14 years. This core assumption can be highly misleading to readers unfamiliar with TRPM2 biology and it is necessary to present credible evidence that TRPM2 is functional in the lysosomal membrane of macrophages. Ideally, this line of investigation should rest on robust demonstration of TRPM2 currents in patch-clamp electrophysiology of lysosomes. If this is not technically feasible for the authors, they should at least investigate TRPM2 localization on lysosomal membranes of macrophages.

In the revised manuscript, authors showed TRPM2 localization but these results are problematic. The authors provide no information on what TRPM2 antibody they used for this study and whether it has been validated by use of knockouts. The staining shows very high amounts of TRPM2 all across the cell - even more than LAMP2. In reality, TRPM2 expression in macrophages is very low. Are the authors overexpressing TRPM2? These data only add to my concerns about this manuscript.

(4) Apigenin and Quercetin are highly non-specific and their effects cannot be attributed to CD38 inhibition alone. Such conclusions need strong loss of function studies using genetic knockouts of CD38 - or at least siRNA knockdown. Importantly, if indeed TRPM2 is being activated downstream of CD38, this should be easily evident in whole cell patch clamp electrophysiology. TRPM2 currents can be resolved using this technique and authors have Trpm2-/- cells for proper controls. Authors attempted these experiments but the results are of very poor quality. If the TRPM2 current is being activated through ADPR generated by CD38 (in response to PA stimulation), then it is very odd that authors need to include 200 uM cADPR to see TRPM2 current (Fig. 3A). Oddly, even these data cast great doubt on the technical quality of the electrophysiology experiments. Even with such high concentrations of cADPr, the TRPM2 current is tiny and Trpm2-/- controls are missing. The current-voltage relationship is not shown, and I feel that the results are merely reporting leak currents seen in measurements with substandard seals. Also 20 uM ACA is not a selective inhibitor of TRPM2 - relying on ACA as the conclusive diagnostic is problematic.

(5) TRPM2 is expressed in many different cell lines. The broad metabolic differences observed by the authors in the Trpm2-/- mice cannot be attributed to macrophage-mediated inflammation. Such a conclusion requires the study of mice wherein Trpm2 is deleted selectively in macrophages or at least in the cells of the myeloid lineage.

(6) The ER-Lysosome Ca2+ refilling experiments rely on transient transfection of organelle-targeted sensors into BMDMs. See point #1 to understand why I find this approach to be highly problematic. Furthermore the data procured are also not convincing and lack critical controls (localization of sensors has not been demonstrated and their response to acute mobilization of Ca2+ has not been shown inspire any confidence in these results).

(7) Authors claim that SCOE is coupled to K+ efflux. But there is no credible evidence that SOCE is activated in PA stimulated macrophages. The data shown in Fig 4 supp 1 do not investigate SOCE in a reliable manner - the conclusion is again based on snapshot measurements and crude non-selective inhibitors. The correct way to evaluate SOCE is to record cytosolic Ca2+ elevations over a period of time in absence and presence of extracellular Ca2+. However, even such recordings can be unreliable since the phenomenon is being investigated hours after PA stimulation. So, the only definitive way to demonstrate that Orai channels are indeed active during this process is through patch clamp electrophysiology of PA stimulated cells.

Authors failed to respond to these concerns in a credible manner and simply tried to obfuscate the matters with extraneous arguments and wild claims. The revised manuscript was not a significant improvement. I have major concerns with this manuscript and let it be on record that this is very poor-quality science.

---

## [Referee Report · Reviewer #2 (Public review)]

In this manuscript by Kang et. al., the authors investigated the mechanisms of K+-efflux-coupled SOCE in NLRP3 inflammasome activation by LP（LPS+PA, and identified an essential role of TRPM2-mediated lysosomal Ca2+ release and subsequent IP3Rs-mediated ER Ca2+ release and store depletion in the process. K+ efflux is shown to be mediated by a Ca2+-activated K+ channel (KCa3.1). LP-induced cytosolic Ca2+ elevation also induced a delayed activation of ASK1 and JNK, leading to ASC oligomerization and NLRP3 inflammasome activation. Overall, this is an interesting and comprehensive study that has identified several novel molecular players in metabolic inflammation. The manuscript can benefit if the following concerns could be addressed.

(1) The expression of TRPM2 in the lysosomes of macrophages needs to more definitively established. For instance, the cADPR-induced TRPM2 currents should be abolished in the TRPM2 KO macrophages. Can you show the lysosomal expression of TRPM2, either with an antibody if available or with a fluorescently-tagged TRPM2 overexpression construct?

In the revised manuscript, the authors did not perform the KO control experiment to support that cADPR-induced currents were indeed mediated by TRPM2. Additonally, the co-localization analyses failed to convincingly establish the lysosomal perimeter membrane residence of TRPM2.

(2) Can you use your TRPM2 inhibitor ACA to pharmacologically phenocopy some results, e.g., about [Ca2+]ER, [Ca2+]LY, and [Ca2+]i from the TRPM2 knockout?

In the revised manuscript, most suggested experiments were not performed. In the only experiment that was conducted, Figure 3-figure supplement 1A, the effect of ACA was marginal.

(3) In Fig. S4A, bathing the cells in zero Ca2+ for three hours might not be ideal. Can you use a SOCE inhibitor, e.g, YM-58483, to make the point?

The specific suggested experiment was not performed.

(4) In Fig. 1A, you need a positive control, e.g., ionomycin, to show that the GPN response was selectively reduced upon LP treatment.

Results in a previous study cannot be used to substitute the missing control experiments in the current study.

---

## [Author Response]

The following is the authors’ response to the original reviews.

**eLife assessment**
The authors present a potentially useful model involving Ca2+ signaling in inflammasome activation. As it stands, it was felt that the data were not sufficient to support the model and the claims of the study are inadequately presented.
**Public Reviews:**

**Reviewer #1 (Public Review):**
This manuscript proposes a complex unclear model involving Ca2+ signaling in inflammasome activation. The experimental approaches used to study the calcium dynamics are problematic and the results shown are of inadequate quality. The major claims of this manuscript are not adequately substantiated.Major concerns:(1) The analysis of lysosomal Ca2+release is being carried out after many hours of treatment. Such evidence is not meaningful to claim that PA activates Ca2+ efflux from lysosome and even if this phenomenon was robust, it is not doubtful that such kinetics are meaningful for the regulation of inflammasome activation. Furthermore, the evidence for lysosomal Ca2+ release is indirect and relies on a convoluted process that doesn't make any conceptual sense to me. In addition to these major shortcomings, the indirect evidence of perilysosomal Ca2+ elevation is also of very poor quality and from the standpoint of my expertise in calcium signaling, the data are incredulous. The use of GCaMP3-ML1, *transiently transfected* into BMDMs is highly problematic. The efficiency of transfection in BMDMs is always extremely low and overexpression of the sensor in a few rare cells can lead to erroneous observations. The overexpression also results in gross mislocalization of such membrane-bound sensors. The accumulation of GCaMP3-ML1 in the ER of these cells would prevent any credible measurements of perilysosomal Ca2+ signals. A meaningful investigation of this process in primary macrophages requires the generation of a mouse line wherein the sensor is expressed at low levels in myeloid cells, and shown to be localized almost exclusively in the lysosomal membrane. The mechanistic framework built around these major conceptual and technical flaws is not especially meaningful and since these are foundational results, I cannot take the main claims of this study seriously.

Ans We agree with the reviewer’s concern that transfection efficiency could be low in BMDMs together with possible mislocalization of GCAMP3-ML1. However, in our experiment, transfection of BMDM with test plasmids resulted in good expression of test proteins. Below, we present our data showing good transfection efficiency of BMDM cells, while a different plasmid was employed.

(2) The cytosolic Ca2+ imaging shown in Figure 1C doesn't make any sense. It looks like a snapshot of basal Ca2+ many hours after PA treatment - calcium elevations are highly dynamic. Snapshot measurements are not helpful and analyses of Calcium dynamics requires a recording over a certain timespan. Unfortunately, this technical approach has been used throughout the manuscript. Also, BAPTA-AM abrogates IL-1b secretion because IL-1b transcription is Ca2+ dependent - the result shown in figure 1D does not shed light on anything to do with inflammasome activation and it is misleading to suggest that.

Ans We agree with the reviewer’s concern that snapshot could lead to false conclusion. We have not traced cytosolic Ca2+ content after treatment with LPS + PA. However, we have tracedlysosomal Ca2+ and ER Ca2+ for more than 15 min, which was presented in Figure 4B. We also agree with the comment that BAPTA-AM might affect transcription of pro-IL-1β. We have conducted immunoblot analysis after treatment with LPS+PA in the presence of BAPTA-AM. Protein band of pro-IL-1β was not affected by BAPTA-AM treatment suggesting no effect of BAPTA-AM on transcription or translation of pro-IL-1β, which was added to Figure 1D, as suggested.

(3) Trpm2-/- macrophages are known to be hyporesponsive to inflammatory stimuli - the reduced secretion of IL-1b by these macrophages is not novel. From a mechanistic perspective, this study does not add much to that observation and the proposed role of TRPM2 as a lysosomal Ca2+ release channel is not substantiated by good quality Ca2+ imaging data (see point 3 above). Furthermore, the study assumes that TRPM2 is a lysosomal ion channel. One paper reported TRPM2 in the lysosomes but this is a controversial claim, with no replication or further development in the last 14 years. This core assumption can be highly misleading to readers unfamiliar with TRPM2 biology and it is necessary to present credible evidence that TRPM2 is functional in the lysosomal membrane of macrophages. Ideally, this line of investigation should rest on robust demonstration of TRPM2 currents in patch-clamp electrophysiology of lysosomes. If this is not technically feasible for the authors, they should at least investigate TRPM2 localization on lysosomal membranes of macrophages.

Ans We agree with the reviewer’s comment that TRPM2. However, we have shown that TRPM2 current was not activated in the plasma membrane of BMDMs after treatment with LPS+PA. We also agree with the reviewer’s comment that inflammatory cytokine release from TRPM2 KO cells or inflammasome response of TRPM2 KO macrophages to ROS or nanoparticles has been reported to be reduced; however, the role of TRPM2 in metabolic inflammation or inflammasome activation in response to lipid stimulators has not been shown, as discussed in the new lines 9-10 from the bottom of page 18. Regarding the role of lysosomal TRPM2 in inflammation, we have shown that bafilomycin A1 treatment abrogated increase of cytosolic Ca2+ by LPS+PA (Figure 3-figure supplement 1D), supporting the role of lysosome and lysosomal Ca2+ in inflammasome activation by LPS+PA.

We agree with the reviewer’s comment that TRPM2 expression on lysosome needs to be tested. We conducted confocal microscopy after immunofluorescence staining using anti-TRMP2 and -LAMP2 antibodies, which showed a certain portion of TRPM2 was colocalized with LAMP-2. This result substantiating TRPM2 expression on lysosome of macrophages was incorporated as Figure 2-figure supplement 1A.

(4) Apigenin and Quercetin are highly non-specific and their effects cannot be attributed to CD38 inhibition alone. Such conclusions need strong loss of function studies using genetic knockouts of CD38 - or at least siRNA knockdown. Importantly, if indeed TRPM2 is being activated downstream of CD38, this should be easily evident in whole cell patch clamp electrophysiology. TRPM2 currents can be resolved using this technique and authors have Trpm2-/- cells for proper controls. Authors attempted these experiments but the results are of very poor quality. If the TRPM2 current is being activated through ADPR generated by CD38 (in response to PA stimulation), then it is very odd that authors need to include 200 uM cADPR to see TRPM2 current (Fig. 3A). Oddly, even these data cast great doubt on the technical quality of the electrophysiology experiments. Even with such high concentrations of cADPr, the TRPM2 current is tiny and Trpm2-/- controls are missing. The current-voltage relationship is not shown, and I feel that the results are merely reporting leak currents seen in measurements with substandard seals. Also 20 uM ACA is not a selective inhibitor of TRPM2 - relying on ACA as the conclusive diagnostic is problematic.

Ans We agree with the reviewer’s comment that effects of apigenin and quercetin could be due to mechanisms other than inhibition of CD38-mediated inflammasome activation. Indeed, that is the reason we have used TRPM2 KO mice and cells. Small TRPM2 current after treatment with high concentrations of cADPr might suggest the minor role of plasma membrane of TRPM2 in macrophage. Regarding concern about ACA, we added data showing inhibition of IL-1β release in response to LPS+PA by ACA as a new Figure 3-figure supplement 1A.

(5) TRPM2 is expressed in many different cell lines. The broad metabolic differences observed by the authors in the Trpm2-/- mice cannot be attributed to macrophage-mediated inflammation. Such a conclusion requires the study of mice wherein Trpm2 is deleted selectively in macrophages or at least in the cells of the myeloid lineage.

Ans We agree with the reviewer’s comment that TRPM2 in cells other than macrophage might have affected the results. Thus, we have conducted in vitro stimulation of TRPM2-KO primary peritoneal macrophages with LPS+PA. We have observed that IL-1β release of TRPM2-KO macrophages in response in vitro treatment with LPS+PA was significantly lower than that from wild-type macrophages (Figure 2C & D), showing the role of TRPM2 in macrophages in inflammasome activation by LPS+PA, which could be independent of TRPM2 in tissues or cells other than macrophages.

(6) The ER-Lysosome Ca2+ refilling experiments rely on transient transfection of organelle-targeted sensors into BMDMs. See point #1 to understand why I find this approach to be highly problematic. Furthermore, the data procured are also not convincing and lack critical controls (localization of sensors has not been demonstrated and their response to acute mobilization of Ca2+ has not been shown to inspire any confidence in these results).

Ans We agree with the reviewer’s comment that transfection or ER-targeted Ca2+ sensor could have artifactual effects. However, we have studied ER-Lysosome Ca2+ experiment using not only GEM-CEPIAer but also using D1ER, a FRET-based ER Ca2+ sensor which has an advantage of short distance of molecular interaction. Thus, we believe that changes of ER Ca2+ after treatment with LPS+PA is not due to an artifactual effect. Multiple contact between VAPA and ORP1L (Figure 4E) also supports ER-lysosome contact, likely facilitating ER-lysosome Ca2+ flux.

(7) Authors claim that SCOE is coupled to K+ efflux. But there is no credible evidence that SOCE is activated in PA stimulated macrophages. The data shown in Fig 4 supp 1 do not investigate SOCE in a reliable manner - the conclusion is again based on snapshot measurements and crude non-selective inhibitors. The correct way to evaluate SOCE is to record cytosolic Ca2+ elevations over a period of time in absence and presence of extracellular Ca2+. However, even such recordings can be unreliable since the phenomenon is being investigated hours after PA stimulation. So, the only definitive way to demonstrate that Orai channels are indeed active during this process is through patch clamp electrophysiology of PA stimulated cells.

Ans We agree with the reviewer’s comment that the final proof of SOCE activation is activation of Orai channel evidenced by electrophysiology. However, we have shown STIM1 aggregation colocalized with Ora1, which is another strong evidence of SOCE channel activation (Vaca L. Cell Calcium 47:199, 2010). Such a paper showing the role of SOCE aggregation in SOCE activation was incorporated in the text (line 4 from the bottom of page 10) and References.

**Reviewer #2 (Public Review):**
In this manuscript by Kang et. al., the authors investigated the mechanisms of K+-efflux-coupled SOCE in NLRP3 inflammasome activation by LP（LPS+PA, and identified an essential role of TRPM2-mediated lysosomal Ca2+ release and subsequent IP3Rs-mediated ER Ca2+ release and store depletion in the process. K+ efflux is shown to be mediated by a Ca2+-activated K+ channel (KCa3.1). LP-induced cytosolic Ca2+ elevation also induced a delayed activation of ASK1 and JNK, leading to ASC oligomerization and NLRP3 inflammasome activation. Overall, this is an interesting and comprehensive study that has identified several novel molecular players in metabolic inflammation. The manuscript can benefit if the following concerns could be addressed:(1) The expression of TRPM2 in the lysosomes of macrophages needs to be more definitively established. For instance, the cADPR-induced TRPM2 currents should be abolished in the TRPM2 KO macrophages. Can you show the lysosomal expression of TRPM2, either with an antibody if available or with a fluorescently-tagged TRPM2 overexpression construct?

Ans We agree with the reviewer’s comment that TRPM2 expression on lysosome needs to be tested. We conducted confocal microscopy after immunofluorescent staining using anti-TRMP2 and -LAMP2 antibodies, which showed a certain portion of TRPM2 was colocalized with LAMP2. This result was incorporated as Figure 2-figure supplement 1A.

(2) Can you use your TRPM2 inhibitor ACA to pharmacologically phenocopy some results, e.g., about [Ca2+]ER, [Ca2+]LY, and [Ca2+]i from the TRPM2 knockout?Ans We agree with the reviewer’s comment that the effect of ACA on other experimental results needs to be shown. We did not study the effect of ACA on Ca2+ flux; however, we have observed that ACA inhibited IL-1β release in response to LPS+PA. This data was incorporated as Figure 3-figure supplement 1A.

**Author response image 2. sa3fig2:** 

(3) In Fig. S4A, bathing the cells in zero Ca2+ for three hours might not be ideal. Can you use a SOCE inhibitor, e.g, YM-58483, to make the point?

Ans We agree with the reviewer’s comment that SOCE inhibitor experiment would be necessary in addition to the experiment employing zero Ca2+. In fact, we have already used two SOCE inhibitors (2-APB and BTP2) (Figure 4-fig. supplement 1 B-D). Particularly, BTP2 experiment could eliminate possible role of ER Ca2+ inhibition that might occur when 2-APB was employed.

(4) In Fig. 1A, you need a positive control, e.g., ionomycin, to show that the GPN response was selectively reduced upon LP treatment.

Ans We did not employ ionomycin as a control in this study. In our previous study using other agents inducing lysosomal Ca2+ efflux, we have observed lysosomal Ca2+ efflux with intact subsequent ionomycin response. While we did not include ionomycin in the current paper, we are positive that ionomycin response would be preserved.

**Recommendations for the authors:**

**Reviewer #1 (Recommendations For The Authors):**
See Public Review.
**Reviewer #2 (Recommendations For The Authors):**
(5) In Fig. 4B, the red label should read "BAPTA-1 Dextran", but not "GAPTA-1 Dextran".(6) Writing should be improved in many sections.